# A Novel Authentication Scheme Based on Verifiable Credentials Using Digital Identity in the Context of Web 3.0

Stefania Loredana Nita [1] and Marius Iulian Mihailescu [2,*]

1    Department of Computers and Cyber Security, Military Technical Academy "Ferdinand I",
     050141 Bucharest, Romania; stefania.nita@outlook.com
2    Scientific Research Center in Mathematics and Computer Science, "Spiru Haret" University of Bucharest,
     030045 Bucharest, Romania
*    Correspondence: m.mihailescu.mi@spiruharet.ro

**Abstract:** This paper explores the concept of digital identity in the evolving landscape of Web 3.0, focusing on the development and implications of a novel authentication scheme using verifiable credentials. The background sets the stage by placing digital identity within the broad context of Web 3.0's decentralized, blockchain-based internet, highlighting the transition from earlier web paradigms. The methods section outlines the theoretical framework and technologies employed, such as blockchain, smart contracts, and cryptographic algorithms. The results summarize the main findings, including the proposed authentication scheme's ability to enhance user control, security, and privacy in digital interactions. Finally, the conclusions discuss the broader implications of this scheme for future online transactions and digital identity management, emphasizing the shift towards self-sovereignty and reduced reliance on centralized authorities.

**Keywords:** digital identity; Web 3.0; blockchain; verifiable credentials; self-sovereign identity





## 1. Introduction

In the context of Web 3.0, the idea of digital identity signifies a fundamental change in the way people and things express and maintain their online presence. Web 3.0 offers an unparalleled level of user control over personal data through a decentralized blockchain-based Internet, marking a departure from the static pages of Web 1.0 and the interactive, social media-driven Web 2.0. This new paradigm aims to provide a more private, secure, and interoperable Internet experience by utilizing technologies like blockchain, smart contracts, and tokenization [1–5].

Web 3.0's concept of digital identity goes beyond the social media accounts and online profiles that typified the preceding Internet era. It includes a comprehensive, user-owned, self-sovereign identity (SSI) that is unaffected by a central authority or middleman. In addition to social connections, this identity is utilized for banking, service access, and engagement in virtual communities and decentralized apps (dApps) [5–8].

A number of significant advantages come with the shift to Web 3.0 digital identities, such as improved user agency, security, and privacy. Personal data are kept private and secure through the use of cryptographic algorithms and decentralized storage, where users choose what information to share and with whom. This paradigm also makes it easier for platforms and services to work together, enabling a seamless digital experience that goes beyond conventionally divided identities [9].

This change, however, also brings with it some difficulties, such as the requirement for new technologies to be widely used, the creation of strong legal and regulatory frameworks to safeguard users, and the possibility of emerging new types of digital inequality. The idea of digital identity in Web 3.0 will surely change as we make this shift, influencing how people connect online, transact, and engage with their communities in the future [10].

The evolution of digital identity and the emergence of Web 3.0 represent significant shifts in how individuals and entities are represented and interact online. Here is a brief overview:

1. Early Digital Identity (Web 1.0 Era)
   - In the early days of the Internet (Web 1.0), digital identity was quite basic and primarily functioned as a means of user identification for email and basic online services [11].
   - Users typically had limited control over their digital identities, which were often tied to specific platforms or services [12].
   - Anonymity was relatively easy due to the lack of interconnected systems and sophisticated tracking [13].

2. Social media and expanded Digital Identity (Web 2.0 Era)
   - With the advent of Web 2.0, characterized by the rise of social media and user-generated content, digital identities have become more complex and multi-faceted [14].
   - Online profiles began to encompass a broader range of personal information, preferences, and social connections [15].
   - This era saw the growth of centralized platforms (like Facebook, Twitter, LinkedIn) that controlled and monetized user data. Privacy concerns and data breaches became more prominent [16].

3. Data ownership and privacy concerns
   - Increasing awareness of data privacy issues led to a push for better control over personal data and digital identity [17].
   - Regulations like GDPR (General Data Protection Regulation) in the EU were introduced to give users more control over their personal data [18].

4. Emergence of Web 3.0 and its impact on digital identity
   - Web 3.0 represents a paradigm shift towards a more decentralized Internet, leveraging technologies like blockchain and peer-to-peer networks [19].
   - In this context, digital identities are evolving towards self-sovereignty, where individuals have greater control and ownership of their digital identities without relying on central authorities [20].
   - Technologies like blockchain enable the creation of secure, verifiable digital identities, enhancing privacy and reducing the risk of identity theft [21].
   - Concepts like Decentralized Identifiers (DIDs) and verifiable credentials are emerging as new standards for self-managed, interoperable digital identities [22].
   - Web 3.0 also brings the potential for interoperable digital identities across different platforms and services without sacrificing user privacy or security [23].

In the context of Web 3.0, digital identities face a variety of challenges and opportunities, each shaping the landscape of how we interact with the digital world. Here is an outline that captures the following challenges and opportunities.

## 1.1. Challenges

The development of digital identity offers both ground-breaking benefits and major obstacles as we enter the Web 3.0 era. With its promise of increased user control, security, and privacy, this new frontier is poised to completely change the way we interact with the Internet. But there are many obstacles to overcome and a difficult road ahead before we can truly have a Web 3.0 digital identity ecosystem. These challenges [24] can be summarized as follows:

1. Technical complexity. The underlying technologies of Web 3.0, like blockchain and decentralized networks, are complex. This complexity can be a barrier to the widespread adoption and understanding of new digital identity systems.

2. Interoperability. Ensuring that different systems and platforms can work together seamlessly is a significant challenge. Digital identities must be universally recognizable and usable across various platforms within the Web 3.0 ecosystem.
3. Privacy and security. While Web 3.0 aims to enhance privacy and security, its decentralized nature also raises new concerns. For instance, the immutable nature of blockchain can be a double-edged sword, making the right to be forgotten (a key aspect of privacy regulations like GDPR) difficult to implement.
4. Regulatory compliance. The decentralized and global nature of Web 3.0 complicates regulatory compliance. Laws and regulations regarding digital identity, privacy, and data protection vary widely across jurisdictions.
5. User adoption and trust. Transitioning from traditional digital identity systems to those based on Web 3.0 technologies requires user trust and willingness to adopt new paradigms. Overcoming skepticism and inertia is a significant challenge.
6. Digital divide. The digital divide could be exacerbated if access to the necessary technology for Web 3.0 digital identities is not universally available. This could lead to further inequality in digital participation.

### 1.2. Opportunities

The dawn of Web 3.0 brings with it a transformative shift in the landscape of digital identity, heralding a future where users gain unprecedented control and security over their personal data. This evolution is not just about technological advancements but represents a reimagining of online identity, offering a plethora of opportunities that could reshape our digital lives. These opportunities [25] can be stated as follows:

1. Self-sovereignty. Web 3.0 offers the opportunity for users to have greater control over their digital identities, known as self-sovereign identity (SSI). This means users can control how their personal information is shared and used [26].
2. Enhanced security and privacy. Technologies like blockchain provide enhanced security features (such as encryption and decentralization) that make digital identities more secure and less prone to fraud or theft [27].
3. Interoperability and portability. Digital identities in Web 3.0 can be designed to be portable and interoperable across multiple platforms and services, offering a more seamless and integrated user experience [28].
4. Innovation in services and applications. The new paradigm of digital identity in Web 3.0 opens up possibilities for innovative applications and services that leverage the enhanced features of these identities, such as improved personalization and decentralized finance (DeFi) [29].
5. Reduced reliance on centralized authorities. By decentralizing identity management, there is less reliance on central authorities, potentially reducing the risk of mass data breaches and the misuse of personal data [30].
6. Inclusion. Properly implemented, Web 3.0 digital identities can increase inclusion, providing identity solutions for individuals who are currently underserved by traditional systems.

### 1.3. Paper Objective and Contributions

The paper aims to discuss the significant changes in how people and organizations create and manage their online presence using digital identities within the framework of Web 3.0. Web 3.0 is a fundamental change from previous web versions, providing users with a substantial degree of control over their personal data. This is accomplished by utilizing a decentralized, blockchain-powered Internet, which differs from the static pages of Web 1.0 and the interactive, social media-focused Web 2.0.

Digital identity in Web 3.0 goes beyond conventional social media accounts and online identities. The system involves a thorough, user-controlled, self-governing identity (SSI) that functions apart from central authority or intermediaries. This type of identification

is used not only for social interactions but also in fields such as banking, service access, participation in virtual communities, and decentralized apps (dApps).

Transitioning to Web 3.0 digital identities offers several advantages, such as enhanced user control, security, and privacy. Personal data are safeguarded via cryptographic methods and decentralized storage, giving users authority over the information they disclose and to whom. This new paradigm promotes collaboration between platforms and services, resulting in a smooth digital experience that goes beyond traditional divisions.

Nevertheless, this transition comes with difficulties, including the requirement of the extensive acceptance of novel technologies, the creation of strong legal and regulatory structures for user safeguarding, and the possible rise of fresh types of digital disparity. In the context of Web 3.0, the development of digital identity is anticipated to have a substantial impact on how individuals interact, conduct transactions, and participate in online communities in the future.

### 1.4. Paper Structure

The structure of the paper includes an introduction to the evolution of digital identity across Web 1.0, 2.0, and 3.0, followed by a detailed literature review. This review delves into theoretical foundations, technological challenges, regulatory and ethical considerations, and user adoption issues related to digital identity in Web 3.0. The paper then presents a framework covering cryptography, decentralized systems, zero-knowledge proofs, and verifiable credentials. The descriptions of the sections are as follows:

- Section 1. Introduction. This section discusses the evolution of digital identity from Web 1.0 to Web 3.0, emphasizing the shift towards a more private, secure, and interoperable Internet experience enabled by technologies like blockchain, smart contracts, and tokenization.
- Section 2. Literature Review. This section provides an overview of the theoretical foundations of self-sovereign identity (SSI), the technical workings of decentralized systems, and the ethical, societal, and regulatory ramifications of a more secure and independent digital identity framework.
- Section 3. Preliminaries and Section 4. Theoretical Framework. Covers the key concepts in cryptography, decentralized systems, zero-knowledge proofs, and verifiable credentials essential for understanding the proposed digital identity model.
- Section 4. The Proposed Verifiable Credentials Authentication Scheme. Details an authentication process that leverages verifiable credentials to confirm and validate a user's digital identity in a secure, user-friendly manner without excessive disclosure of personal information.
- Section 5. Security Analysis. This section analyses the security aspects of the proposed system, focusing on zero-knowledge proofs (ZKPs) and their properties like completeness, soundness, and zero-knowledge to ensure secure digital transactions.

Each component helps develop a thorough comprehension of digital identity in the context of Web 3.0, emphasizing improved security, privacy, and user autonomy.

## 2. Literature Review

The idea of digital identity in the context of Web 3.0 has become a crucial topic of scholarly investigation in the quickly changing field of digital technology. The theoretical foundations of self-sovereign identity (SSI), the technical workings of decentralized systems, and the intricate web of ethical, societal, and regulatory ramifications that come with the shift to a more secure and independent digital identity framework are just a few of the many subjects covered in this corpus of the literature. The literature review provides a thorough overview by combining important themes, arguments, and insights from a range of sources, helping academics and practitioners as they wrestle with the opportunities and problems brought about by this new paradigm.

The review starts out by defining key terms and concepts that are essential to the discussion of digital identity in Web 3.0, such as SSI and the technical framework supporting

decentralized identities. This section emphasizes the significance of blockchain technology in enabling safe and private online interactions, laying the foundation for understanding the move towards a more user-centric model of identity management.

The literature then discusses the many technical issues, including scalability, interoperability, and striking a balance between privacy and security, that arise when implementing these systems at scale. Scholars suggest inventive methods and procedures to surmount these obstacles, emphasizing the continuous advancement and enhancement of Web 3.0 technology.

Another important aspect of the literature is the analysis of regulatory and ethical issues, as researchers examine how decentralized digital identities affect adherence to international data protection regulations and ethical norms. These conversations highlight the necessity of a sophisticated approach to regulation and governance that respects user rights and privacy while being in line with the decentralized principles of Web 3.0.

Other major issues include user adoption and the impact on society, with research examining the variables that affect people's faith in and acceptance of new digital identification systems. The body of research highlights the difficulties associated with the digital divide and fair access, as well as the possible advantages of Web 3.0 digital identities, such as enhanced inclusion and creativity across numerous industries.

Future directions and emerging trends in the subject are also explored, with special emphasis on how digital identities might be integrated with other cutting-edge technologies like artificial intelligence (AI) and the Internet of Things (IoT). While raising concerns about blockchain's possible negative effects on the environment and the need for sustainable solutions, researchers also conjecture about the potentially revolutionary nature of these convergences.

All things considered, the literature review (see Table 1) for the current work provides a thorough and comprehensive examination of the social, ethical, legal, and technological aspects of digital identities. This study highlights the difficult problems and intriguing opportunities that lie ahead while also capturing the status of the subject through the integration of multiple viewpoints and findings.

**Table 1.** Literature review.

| Category of Reviews | Works | Source |
|---|---|---|
| Theoretical foundations and definitions | [31] | MDPI Future Internet |
| | [32–40] | IEEE |
| | [41–47] | Springer |
| | [48,49] | Packt |
| Technological challenges and solutions | [50–70] | IEEE |
| | [71–87] | MDPI Future Internet, MDPI Cryptography |
| | [88–103] | Springer |
| Regulatory and ethical considerations | [104–124] | MDPI Cryptography, Future Internet, Electronics |
| | [125–138] | IEEE |
| | [139–152] | Springer |
| User adoption and societal impact | [153–165] | IEEE |
| Future directions and emerging trends | [166–179] | IEEE |

The works from the Theoretical foundations and definitions category (see Table 1) include a wide range of studies and conversations about the relationship between Web 3.0 and digital identity, providing information about the theoretical foundations and real-world applications of these ideas in a variety of fields. In their investigation of Web 3.0's effects on the media industry, ref. [31] emphasize how decentralized technologies have the ability

to revolutionize the production and dissemination of information. In [32], the authors highlight the fundamental components of Web 3.0's future Internet by emphasizing the significance of blockchain in building decentralized trust and secure transactions. In their exploration of the combination of Web 3.0 and e-learning, the [33] highlight the advantages and disadvantages of implementing these new technologies in education.

In addition to delivering a deep dive into the technological details that drive Web 3.0, Bashir's work from [48] gives a thorough overview of the inner workings of blockchain, including its application in digital identities, decentralized finance (DeFi), and non-fungible tokens (NFTs). The social ramifications of Web 3.0 technologies are discussed by [41,42], who concentrate on questions of freedom, social inclusion, and the critical elements influencing decentralized web layers in business and industry. For a better understanding of the blockchain and trustworthy systems in the context of education and information technology, consult the encyclopedic works of Tatnall [43] and Dai [44].

In [49], there is a thorough manual that provides a close examination of all the elements that make up the blockchain ecosystem. The work goes into great detail about the inner workings of blockchain technology, covering everything from consensus processes and distributed ledgers to smart contracts and decentralized apps (DApps), which are more advanced subjects. In-depth explanations of these technologies' functions and interconnections within the larger context of online interactions and digital transactions are given by Bashir.

The works [45,46] add to the conversation about digital transformation and social network analysis, respectively, emphasizing the sustainable and societal elements of the advancement of digital technology. Early research on Web 3.0 and its implications for e-learning, search engines, and the Semantic Web as a whole is traced by [34,35].

In [36], the authors discuss the development of a platform for creating context-aware systems by domain experts, emphasizing the importance of context in the emerging Web 3.0 environment. This work is significant for digital identity as it highlights the potential for leveraging contextual data to enhance the security, personalization, and relevance of digital interactions. By enabling non-experts to contribute to the development of context-aware applications, this research points towards a more inclusive and adaptable Web 3.0 ecosystem.

Work [47] acts as a vital resource in the area of security and cryptography, providing thorough explanations of important ideas, methods, and applications. This encyclopedia covers subjects including encryption, digital signatures, and public key infrastructure (PKI) and discusses the crucial role that cryptographic concepts play in protecting digital identities inside Web 3.0. Because Web 3.0 is decentralized, it is essential to comprehend these cryptographic foundations to create digital identification solutions that are reliable and safe.

Paper [37] clears the collaborative and semantic elements of the next-generation web by examining the management of common knowledge within the framework of Web 3.0. Insofar as it discusses the difficulties and possibilities involved in producing, disseminating, and applying communal knowledge in a decentralized digital environment, this research is pertinent to digital identity. Potentially more dynamic, knowledge-driven identification systems that use collective intelligence for reputation management, authentication, and trust-building are among the implications for digital identity.

Paper [38] is essential to creating the foundation for Web 3.0, also known as the Semantic Web. This essay explores the idea of a web that is both human-readable and computer-interpretable, allowing computers to carry out increasingly complex functions on behalf of users. The focus on "embracing Web 3.0" draws attention to the evolution of the web into a more intelligent, networked environment that makes unprecedented use of data. This evolution points to a shift in digital identification toward identities that are self-sovereign and able to engage in a semantic, interconnected framework where services and data are effortlessly integrated.

In [39], the authors present the concept of weighted myriad filters, a robust filtering framework derived from alpha-stable distributions. This research contributes to the broader discussion of data processing and management techniques that are essential in the context of Web 3.0, especially when dealing with large volumes of data or noisy environments. For digital identity, the relevance lies in the potential application of such advanced filtering techniques to enhance the security, privacy, and reliability of identity data as they are processed and analyzed across decentralized networks.

In contrast to the central themes of Web 3.0 and digital identity, the study [40] explores the field of signal processing. This study presents a complex filtering technique that is intended to function well in settings with impulsive noise, which is typical of many signal processing applications. The paper adds to the wider debate on data processing and analysis methods by concentrating on numerous weighted filters and how they manage non-Gaussian noise and outliers in signal data. Gonzalez and Arce present a technique that improves the durability and reliability of filtering processes by utilizing alpha-stable distributions. This guarantees that the resulting data are more accurate and cleaner, even under difficult noise situations.

The collection of papers and works spans a wide range of topics within the fields of digital design, cybersecurity, blockchain technology, cloud computing, and the Internet of Things (IoT), reflecting the current trends and challenges in information technology and engineering. In [50], the necessity of a holistic design approach to shape a sustainable digital future is described, arguing for the integration of sustainability principles into the digital design process. This is pivotal for guiding future practices towards more sustainable outcomes. In [51], the authors address evolving threats in modern hybrid cloud architectures, presenting AI-driven solutions to enhance security and resilience and showcasing the potential of AI in fortifying cloud infrastructures. The [52] provides an exploration of blockchain technology in HRM. In [53], the authors discuss how blockchain can enhance government operations' transparency and efficiency, leveraging its immutability and decentralization. In [54], the authors delve into the specific challenges of implementing blockchain within the telecom sector, suggesting pathways to harness its benefits for enhancing security and service delivery. In [55], the authors focus on the paramount importance of cloud security, proposing comprehensive strategies for ensuring data protection. In [56], the authors introduce a model for a digital platform aimed at combating cyberaggression in educational settings, contributing to safer digital spaces. In [57], the authors present a case study on the challenges of digitalization in a remote island setting, offering insights into digital inclusion and accessibility. In [58], the authors explore blockchain design patterns to address common challenges, providing valuable insights for developers. The paper from [59] focuses on cybersecurity for small enterprises and is significant for making cybersecurity accessible and manageable for small businesses. The authors propose in [60] a blockchain framework to secure digital identity transactions in the Indian agri-subsidy system, highlighting blockchain's potential in public sector applications. In [61], the authors focus on IoT security requirements, identify key issues, and propose solutions to enhance IoT security. In [62], Keung et al.'s case study on cloud-based cyber–physical robotic systems showcase the integration of cloud computing with physical robotics systems. In [63], the authors discuss a secure, self-sovereign identity framework for IoT devices, addressing identity management challenges. In [64], the authors introduce a framework for vehicle-to-vehicle communication, enhancing communication efficiency in vehicular networks. The work in [65] evaluates blockchain-based identity management, highlighting its potential to revolutionize identity verification processes. Ref. [66] explores the security risks associated with IMEIs, emphasizing the need for robust mobile security mechanisms. The work in [67] proposes a blockchain-based design framework for the Indonesian tertiary education system, enhancing educational administration's transparency and efficiency. In [68], the authors present a secure cloud computing framework for smart grid information management, addressing the need for robust security measures. Ref. [69] proposes a privacy-aware incentive mechanism for mobile data collection, addressing the

need for privacy protection. Paper [70] gives a presentation on cyber warfare and terrorism and calls for increased awareness and preparedness to counteract cyber threats. Collectively, these works underscore the multifaceted challenges and opportunities presented by advances in digital technologies, contributing valuable insights and solutions critical for advancing technology in a secure, efficient, and beneficial manner for society.

The field of digital technology is dynamic and ever-evolving, as evidenced by the recent explosion of research in several domains such as machine learning, WebAssembly, digital competency management, blockchain, web-based systems, machine learning, and cyber–physical systems. Ref. [71] Ferilli et al.'s investigation into a graph database-based approach for semantic technologies in the Internet of the future presents a viable path for improving data interpretation and connectivity, which is essential for the development of the semantic web. [72] A major contribution to our understanding of blockchain's financial features is made by Marin et al., who offer a thorough analysis of blockchain tokens and throw light on their formation, pricing, and wider implications for digital economies. In Ref. [73], an innovative method for assessing user involvement and system efficiency—both vital for enhancing web interfaces—is presented by AlSalem and AlShamari's evaluation of interactive web-based systems using behavioral measurement techniques. In Ref. [74], Web 3.0 environments, Francia et al.'s discussion on digital competency management using the C-Box® approach emphasizes the value of cutting-edge tools for managing and certifying digital skills, reflecting the move toward more independent and customized learning and professional development platforms.

In Ref. [75], Bucur and Miclea's analysis of using JVM-based tools and libraries to access the Metaverse tackles the research topics and technological difficulties while providing insights into the creation of dynamic, immersive virtual worlds. In Ref. [76], Fragiadakis et al.'s use of machine learning to forecast the costs of cloud services, particularly Amazon IaaS, shows how AI has the ability to lower costs and increase accessibility to cloud computing. In Ref. [77], Ray's summary of WebAssembly for IoT captures the technology's potential to improve the security and performance of IoT applications, signaling a move toward more reliable and effective web technologies. In Ref. [78], the pragmatic web and the web of social representations, as envisioned by Haralambous and Lenca, expand on the semantic web's bounds and offer a more sophisticated comprehension of web material and its social ramifications.

In Ref. [79], the creation of a cyber–physical system by Battistoni et al. for the detection and fighting of wildfires is an example of how digital and physical systems can be integrated to handle important environmental concerns. In Ref. [80], Sufi made significant progress in cybersecurity with the release of a new AI-based semantic cyber intelligence agent that offers enhanced threat detection and information security features. Ref. [81], tracing the development of non-fungible tokens, presents Guidi and Michienzi's review from NFT 1.0 to NFT 2.0, emphasizing their expanding influence on digital ownership and the creative economy. In Ref. [82], Taherdoost highlights how blockchain technology has the potential to completely transform healthcare information management by guaranteeing interoperability, privacy, and security in his explanation of the technology's role in medical data exchange.

Ref. [83] Bespalov et al.'s methods for the production of proof forests in zk-SNARK-based sidechains address some of the fundamental issues with blockchain technology and improve the scalability and privacy of blockchain applications. In Ref. [84], Ciampi, Romano, and Schmid's analysis of blockchain-based process authentication through three case studies demonstrates the technology's adaptability in guaranteeing process reliability and integrity. In Ref. [85], the use of a blockchain framework in Perera et al.'s certificate management method for VANETs highlights the crucial role that blockchain plays in improving the security and dependability of vehicular communication networks. In Ref. [86], Nelaturu, Du, and Le's study of blockchain in fintech highlights the technology's disruptive potential in financial services by outlining the taxonomy, difficulties, and future possibilities. In Ref. [87], it was found that an essential component of cloud security is the complexity of

managing cryptographic keys in cloud environments, which is covered in Campagna and Gueron's talk [180] on key management systems at the cloud scale.

The development of digital societies and smart cities in the future will be greatly influenced by the latest developments in computing architectures, blockchain technology, and cybersecurity. Numerous academic publications that explore different facets of technology's role in improving security, efficiency, and sustainability in our increasingly interconnected society provide a thorough analysis of these trends.

In Ref. [88], Cano M, J.J. examines the complex angles of cybersecurity risk management, stressing the significance of comprehending the perspectives of both the adversary and the victim. Given the increasing sophistication of digital threats in the era of smart societies, this dual perspective is essential for creating cybersecurity tactics that are more effective.

In Ref. [89], in order to address the issue of high cybersecurity analyst turnover, Adetoye and Fong suggest a multidisciplinary strategy for developing a workforce that is resilient in the field. Their research emphasizes the necessity of an all-encompassing approach that deals with the underlying reasons for employee attrition and cultivates a more reliable and competent workforce.

In Ref. [90], IoT-Penn, a security penetration testing tool created especially for the MQTT protocol in IoT contexts, is introduced by Roets and Tait. The security of Internet of Things (IoT) networks and devices, which are quickly becoming the foundation of smart city infrastructures, advanced significantly thanks to this solution.

In Ref. [91], with a case study on Russian hackers, Ehiorobo, Pournouri, Ghazaani, and Toms concentrate on using classification techniques to profile cyber attackers. Their study adds to the larger endeavor of comprehending the actions of cybercriminals, which is necessary for creating focused cybersecurity defenses.

In Ref. [92], Aranda-Tyrankiewicz and Jahankhani talk about how blockchain technology could be used to stop fake news from spreading on messaging apps and social media. In today's information-rich environment, their study emphasizes the importance of blockchain in boosting digital trust and information integrity.

In Ref. [93], Nyarko and Fong investigate cybersecurity compliance among remote employees, a topic that is relevant given the rise in remote employment. Their research sheds light on the difficulties and solutions associated with preserving cybersecurity in decentralized work settings.

In Ref. [94], Montasari and Boon examine the difficulties that digital policing faces due to the dark web. Global security services are becoming increasingly concerned about the challenges of navigating and implementing legislation in the shadowy regions of the Internet. Their study illuminates these challenges.

In Ref. [95], an IoT case study on edge and fog computing with an emphasis on citizen well-being is presented by Bianconi et al. Their research demonstrates how various computing paradigms may analyze data closer to the source, improving the responsiveness and efficiency of applications for smart cities.

In Ref. [96], Böhm and Wirtz discuss the opportunities and the difficulties of managing distributed computing resources as they investigate the orchestration of cloud-edge systems using Kubernetes. The creation of scalable and robust smart city infrastructures depends heavily on this study.

In Ref. [97], Shen, Zhou, Xie, Yu, and Xuan use a graph neural network to study identity inference on blockchain, providing a novel method for comprehending and protecting digital identities on blockchain networks. The field of blockchain security and privacy is enhanced by this effort.

In Ref. [98], Zhang et al. use unsupervised learning techniques to investigate the regional clustering impact of the blockchain sector, offering insights into the economic and geographic trends influencing the blockchain ecosystem. Both investors and governments should take note of the findings of this study.

In Ref. [99], Gu, Lin, Zheng, Wu, and Hu demonstrate how AI may improve the predictability and stability of blockchain networks by using deep learning to forecast Ethereum

transactions. This strategy has the potential to completely change how decentralized finance manages and predicts transactions.

In Ref. [100], Blockchain aberrant transaction behavior analysis is covered in detail by Han, Chen, Guo, and Zhang, who provide a thorough overview of the difficulties and strategies involved in identifying and comprehending anomalous activity on blockchain networks. The security and integrity of blockchain systems depend on this effort.

In Ref. [101], Zhang, Xu, Dong, and Lin highlight the technology's potential to unify and protect digital identities across many platforms and networks as they address the application and problems of blockchain in heterogeneous identity trust. A basic problem in digital identity management is addressed by this study.

In Ref. [102], Yu, Jin, Xie, Shen, and Xuan concentrate on identifying Ponzi schemes within the Ethereum transaction network, making a valuable contribution to the continuous endeavor to protect investors and uphold confidence in financial systems based on blockchain. Their efforts are a vital first step in the fight against financial fraud in the digital era.

A graph convolutional network is used by Shen, Sang, Duan, Yu, and Zhu [103] to anticipate transaction anomalies in blockchain digital money. This novel method makes use of artificial intelligence to improve the security of digital currencies, which are a crucial component of the financial system of the future.

Collectively, these works show how technology and society interact dynamically while emphasizing the vital roles that cybersecurity and advanced computing play in creating reliable, dependable, and efficient digital ecosystems.

A lively investigation of blockchain technology, privacy-preserving methods, and creative computing solutions targeted at improving data security, management, and compliance across multiple industries can be found in the recent literature in the electronics, future Internet, and cryptography domains. These studies show that there is an increasing focus on safeguarding digital transactions, empowering data subjects, and using blockchain to advance society.

In Ref. [104], Khalid, Ahmed, Helfert, and Kim stress decentralized data controllers and privacy-preserving methods in their privacy-first paradigm for dynamic consent management systems. Giving people more control over their personal data is the goal of this strategy, which is an important first step in improving privacy in digital ecosystems.

In Ref. [105], the use of blockchain technology in healthcare game management is examined by Chen, Cao, and Cai. They show how blockchain can simplify and secure the administration of gamification procedures related to health. This creative use case demonstrates how blockchain technology may be applied to the health and wellness sectors in addition to typical financial applications.

In Ref. [106], Marengo and Pagano perform a thorough literature analysis to look into the variables affecting the adoption of blockchain technology in various nations and sectors of the economy. Their conclusions give stakeholders thinking about implementing blockchain technology useful insights into the obstacles and facilitators of blockchain adoption.

In Ref. [107], Ma, Yu, Du, Li, Ni, and Lv suggest an incentive system for exchanging cyber threat intelligence based on blockchain technology. By promoting cooperation and information exchange amongst stakeholders, this method seeks to strengthen group cybersecurity defenses in an environment lacking in trust.

In Ref. [108], Duan, Wang, Zhang, Ma, and Luo address the need for adaptable yet secure data management within blockchain networks, especially in Internet of Things applications where data integrity and privacy are crucial. They present a policy-based chameleon hash with black-box traceability for redactable blockchain in IoT.

In Ref. [109], Fekete and Kiss investigate the possibilities of utilizing the zero-knowledge Ethereum Virtual Machine to create smart contract-based higher education platforms. With the use of blockchain technology, this method seeks to transform the verification and administration of educational credentials while guaranteeing security and privacy.

In Ref. [110], a blockchain-based GDPR-compliant data storage and sharing solution for smart healthcare systems is presented by Bai, Kumar, Kumar, Kaiwartya, Mahmud, and Lloret. Their approach makes use of blockchain's built-in security characteristics to satisfy the crucial needs of privacy and compliance in the administration of healthcare data.

In Ref. [111], Mahmood and Jusas create a multi-layered security federated learning platform with blockchain functionality to protect user privacy. This technology enables collaborative model training without jeopardizing data privacy, marking a significant development in secure and privacy-preserving machine learning.

In Ref. [112], Humayun, Jhanjhi, Niazi, Amsaad, and Masood emphasize the use of blockchain technology to protect medication distribution networks against manipulation, emphasizing the technology's potential to improve supply chain integrity and safety in the pharmaceutical sector.

In Ref. [113], in an effort to safeguard electronic medical records in hospitals, Hang, Choi, and Kim present a unique EMR integrity management system built on a medical blockchain platform. This system emphasizes how important blockchain is to protecting private health data.

In Ref. [114], Martins Gonçalves, Mira da Silva, and Rupino da Cunha talk about how to use blockchain to condcut GDPR-compliant surveys, demonstrating how useful it is for guaranteeing data protection compliance in survey research.

In Ref. [115], Abd Ali, Yusoff, and Hasan offer a thorough analysis of the redactable blockchain, addressing its workings, difficulties, unresolved problems, and potential future study areas. This paper demonstrates how blockchain technology is developing and how it may be used for flexible data management.

In Ref. [116], to highlight the potential of blockchain data for perceptive analysis and decision-making, Vinceslas, Dogan, Sundareshwar, and Kondoz concentrate on abstracting data in distributed ledger systems for higher-level analytics and visualizations.

In Ref. [117], Cocco, Tonelli, and Marchesi suggest a blockchain, IoT, SSI, and BIM-based information management system for the building industry. This plan embodies a comprehensive strategy for incorporating state-of-the-art technologies to improve transparency and efficiency in the construction sector.

In Ref. [118], blockchain and self-sovereign identity are investigated as ways to improve food supply chain quality in another study by Cocco, Tonelli, and Marchesi. This study emphasizes blockchain's function in guaranteeing traceability and integrity from farm to table.

In Ref. [119], Tjoa, König, Korobeinikova, and Kieseberg evaluate existing frameworks and their consequences for the adoption and standardization of blockchain technology by comparing blockchain standards and recommendations.

In Ref. [120], O'Donovan and O'Sullivan provide a methodical examination of actual energy blockchain projects, providing insights on how blockchain might be used to improve the sustainability and efficiency of energy distribution and consumption.

In Ref. [121], in their exploration of consortium blockchain smart contracts for musical rights governance, Kapsoulis, Psychas, Palaiokrassas, Marinakis, Litke, Varvarigou, Bouchlis, Raouzaiou, Calvo, and Escudero Subirana show how blockchain technology has the potential to completely transform rights management in the creative industries.

In their discussion of linked blockchain federations for the exchange of electronic health records, Hashim, Shuaib, and Sallabi [122] highlight the potential of blockchain technology to safely and effectively handle health data across many stakeholders.

In Ref. [123], Bazydło, Wiśniewski, and Kozdrój demonstrate how blockchain technology may be used to guarantee the lifespan and integrity of digital documents by presenting a reliable and secure durable medium electronic service.

In Ref. [124], Joshi and Banerjee investigate the use of policy-integrated blockchain to automate privacy compliance, demonstrating how blockchain can help with privacy regulation compliance by automatically enforcing policies.

Generally, these studies highlight how blockchain and privacy-preserving technology can revolutionize several industries, including supply chain management, energy, healthcare, and education. By highlighting the continuous efforts to use technology to create more transparent, efficient, and safe systems, they greatly add to the conversation about digital innovation and its effects on society.

The recent upsurge in research on the intersection of blockchain and AI across a range of areas highlights a paradigm shift toward the development of more transparent, efficient, and safe systems. These research works, which were presented at international conferences and published in respectable journals, demonstrate creative ways to use technology to improve data privacy, cross-border data sharing, energy markets, healthcare, and the governance of smart cities.

In Ref. [125], Tripathi and Mishra investigate how blockchain technology and artificial intelligence may work together to control algorithms, and they offer a framework that improves accountability and transparency in automated decision-making. This study highlights how blockchain technology and artificial intelligence can be used to address moral and legal issues with technology adoption.

In Ref. [126], Peng, Sun, Zhou, Zhang, Cui, and others concentrate on improving cross-border data exchange in blockchain networks by using a compliance-centric methodology that guarantees traceability and anonymity. Their work tackles important issues with international data transmission and offers solutions that strike a compromise between privacy and legal compliance.

In Ref. [127], with their blockchain-based data storage system for the power market, Zhao, Chen, Zhang, Wang, and Zou demonstrate how blockchain technology may transform energy trade and distribution while improving security and transparency.

In Ref. [128], Jiang, Zha, Fang, and Yin create a multidimensional parameter credit-based blockchain consensus algorithm, presenting a brand-new method for reaching agreement in decentralized networks. The purpose of this protocol is to increase the efficiency and scalability of blockchain systems.

In Ref. [129], A blockchain-based protocol called Sec-Health is proposed by Costa, Pinheiro, Cordeiro, Araújo, and Abelém to secure health records. Their approach highlights how crucial blockchain is to safeguarding private patient data, data integrity, and sensitive health information.

In Ref. [130], in his discussion of the Cloud of Things and blockchain, Mishra lays out an architecture that blends the security aspects of blockchain technology with the enormous data-generating capacity of IoT devices. The issues of interoperability, privacy, and scalability in Internet of Things applications are addressed by this integration.

In Ref. [131], Yue and Shyu examine how distributed intelligent healthcare based on blockchain technology is evolving from a policy analysis standpoint. Their study focuses on enhancing patient care and service delivery while highlighting the policy implications of implementing blockchain in the healthcare industry.

In Ref. [132], Zhang et al. provide PACTA, a blockchain-based Trusted Execution Environment (TEE)-based solution for Internet of Things data privacy legislation compliance. This strategy seeks to protect data in IoT environments while adhering to strict privacy laws.

In Ref. [133], Chang, Zhai, Han, and Meng highlight the potential of blockchain technology to improve supervision and transparency in industrial operations, and they present a blockchain-based approach to monitoring industrial Internet of Things (IIoT) businesses.

In Ref. [134], in their analysis of the blockchain's potential impact on data privacy in HRM, Rashmi, Sood, Prashar, Shravan, Sivaprasad, and Lourens propose that blockchain technology might be used to create transparent and safe HR procedures.

In Ref. [135], Puri, Solanki, Kataria, and Long talk about a blockchain-enabled smart city regulatory system, highlighting how technology may promote transparent and reliable governing structures that improve urban living.

Wang, Wan, Hu, Yuan, and Fan [136] investigate a cross-chain supervision mechanism for consortium blockchain, tackling the problems of trust and interoperability in contexts with several blockchains. The goal of this technique is to make cross-chain transactions safe and effective.

In Ref. [137], Xu, Tian, Gao, Lei, Liu, and Liu offer an extensive analysis of the use of blockchain technology in supply chain management for pharmaceuticals. Their research highlights how blockchain technology can be used to prevent counterfeiting and protect the integrity of the pharmaceutical supply chain.

In response to the urgent requirement to safeguard private medical picture data from tampering and unwanted access, Lin, Li, Lin, and Tsai [138] suggest a blockchain-based secure storage system.

Generally, these studies demonstrate the wide range of applications and revolutionary possibilities of blockchain and artificial intelligence technology across diverse industries. They add a great deal to the conversation about digital innovation and its effects on society by highlighting continuous efforts to use these technologies to create systems that are transparent, safe, and efficient, in addition to meeting legal requirements.

This new collection of academic publications offers a thorough examination of the ways in which blockchain technology can be applied in a number of industries, such as agribusiness, education, healthcare, and small- and medium-sized businesses (SMEs), in addition to the critical roles that cybersecurity and artificial intelligence play in advancing digitalization and business intelligence. These studies, which were published by Springer, explore the revolutionary possibilities of blockchain technology and artificial intelligence (AI), showcasing creative answers to persistent problems with data protection, management, and applications tailored to industries.

In Refs. [139–143], the series of contributions by Ramasamy and Khan to "Blockchain for Global Education" addresses the use of blockchain to develop a digital identity system for students, introduce blockchain technology into education, and create a decentralized database of educational credentials. It also discusses how to transform education through an e-learning platform based on blockchain. Together, these pieces highlight how blockchain technology might transform the education industry by facilitating accessibility, trust, and openness in the management of identities and credentials for education.

In response to the pressing need for the safe and effective administration of digital identities—a necessity in the age of digital transformation—Satybaldy, Subedi, and Idrees [144] offer a decentralized key management solution for digital identity wallets.

In Ref. [145], by examining smart contract vulnerabilities, Dhillon, Diksha, and Mehrotra illuminate the technological and financial factors that support the security and effectiveness of blockchain applications. Understanding and reducing the risks related to the deployment of smart contracts depends on this study.

In Ref. [146], Mourya, Kapil, and Idrees combine blockchain technology with mobile agents to offer a cutting-edge method of managing healthcare data. The objective of this amalgamation is to augment the confidentiality, integrity, and compatibility of medical records, signifying a noteworthy progression in digital health remedies.

In Ref. [147], Ganeshkumar, David, and Sankar examine the use of blockchain technology in the agribusiness sector, stressing the elements influencing its adoption as well as how it might improve the efficiency and transparency of agricultural supply chains.

In Ref. [148], Kondala, Nudurupati, and Nihar investigate how SMEs are utilizing blockchain technology and circular economy principles, offering insights into how blockchain might assist environmentally friendly business practices and improve small enterprises' operational efficiency.

In papers [149–152], the authors bring significant contributions to "Cyber Security Impact on Digitalization and Business Intelligence" by discussing the influence of supply chain resilience on competitiveness and the impact of cybersecurity strategy on e-logistics performance using empirical data from the electronics and petroleum industries in the United Arab Emirates. Furthermore, they investigate the use of explainable artificial

intelligence (EAI) in HRM systems and disease prediction models, highlighting EAI's ability to improve operational effectiveness and decision-making processes.

When taken as a whole, these pieces show how cutting-edge technology, such as blockchain and artificial intelligence, interact with one another and with applications in other fields. They draw attention to continuing initiatives to use these technologies to build systems that are transparent, compatible with regulations, and more secure and efficient than before. This collection offers insightful information to academics, business experts, and policymakers alike, greatly advancing the conversation on digital innovation and its effects on society.

A recent compilation of papers from multiple international conferences in 2023 demonstrates the creative use of blockchain technology in a variety of industries, including cybersecurity, IoT, education, healthcare, and energy. These studies demonstrate how blockchain technology can revolutionize digital systems by improving security, transparency, and interoperability.

In Ref. [153], Maruthi, Piriadarshani, Padmanabhan, and Shanthi address the crucial requirement for the safe and effective management of IoT devices and data by proposing a secure framework for the application of blockchain technology in IoT. This framework seeks to defend IoT networks from a range of vulnerabilities by utilizing blockchain's built-in security capabilities.

In Ref. [154], Alar, Shuaib, Khormi, Alam, Aqeel, and Ahmad investigate how blockchain-based systems can improve academic records' transparency and trustworthiness. Their research highlights how blockchain technology has the ability to completely transform the worldwide management and verification of educational qualifications.

In Ref. [155], Lenrow is a peer-to-peer blockchain-based product lending and borrowing system that was introduced by Mishra, Kumar, Pal, and Trivedi. This creative application exemplifies how blockchain technology may support safe and untrustworthy sharing economy transactions.

In Ref. [156], Kaushal and Kumar talk about using Hyperledger Fabric to create blockchain technology as well as a method for evaluating performance. Their research offers practical advice for implementing blockchain technology in business environments, emphasizing efficiency, security, and scalability.

In Ref. [157], Yue and Shyu's study, which examines the possibilities and difficulties of incorporating blockchain technology into healthcare systems to enhance patient care and service delivery, sheds light on the creation of a distributed intelligent healthcare industry based on blockchain technology from the standpoint of policy analysis.

In Ref. [158], building a cross-chain identity using a self-sovereign identity-based framework is the focus of Zecchini, Sober, Schulte, and Vitaletti's work. This work offers a solution for easy and safe identity verification, addressing the crucial problem of identity management across various blockchain networks.

In order to guarantee the reliability of professional and academic qualifications, Balamurugan and Sahayaraj [159] present a blockchain-based certificate authentication system. With the implementation of this system, fraud will be eradicated, and certification integrity will be improved.

In Ref. [160], a conceptual understanding of how to achieve interoperability amongst diverse blockchain-enabled interconnected smart microgrids is provided by Dinesha and Patil. Their research examines how blockchain technology might help with energy distribution and trading among various microgrid networks.

In Ref. [161], Alandjani looks at how blockchain technology might affect several businesses. He gives a general idea of how blockchain can transform a number of industries by guaranteeing data integrity, boosting security, and encouraging creativity.

In Ref. [162], Singh, Pant, Kansal, Singh, Singh, and Jauhari explore cybersecurity in the context of blockchain technology, discussing how it can strengthen cybersecurity defenses and fend off new online dangers.

In Ref. [163], Dhawan, Rastogi, and Saisanthiya talk about how blockchain-based record-keeping can revolutionize the healthcare industry. They highlight how blockchain can secure patient data and improve healthcare results by enhancing data management and accessibility.

In Ref. [164], to reduce fraud and boost transparency in Indonesia's presidential election, Inayatulloh, Hartono, and Kusumastuti offer a blockchain conceptual model. This model demonstrates how blockchain technology can be used to guarantee the transparency and integrity of election procedures.

In Ref. [165], Hire, Lanjewar, Haridas, Jadhav, and Rane investigate the idea of a blockchain-based decentralized lottery, demonstrating how blockchain technology can be used to build transparent and equitable lottery systems.

Generally, these works demonstrate the extensive and significant uses of blockchain technology in various fields, underscoring its potential to revolutionize businesses through improved security, efficiency, and transparency. The research that is being presented adds a great deal to the conversation about digital innovation and its effects on society by offering insightful information on the opportunities and difficulties associated with implementing blockchain technology.

The exponential increase in academic publications in a variety of IEEE conferences and journals highlights the revolutionary influence of blockchain technology in a number of fields, such as cybersecurity, industry 4.0, smart cities, healthcare, and vehicle networks. All of these examples demonstrate how blockchain technology may be creatively applied to improve digital system trust, privacy, interoperability, and data security.

In Ref. [166], Arbabi, Lal, Veeraragavan, Marijan, Nygård, and Vitenberg offer an in-depth analysis of blockchain technology in healthcare, covering its advantages, disadvantages, and potential uses. The promise of blockchain technology to transform healthcare data management through security, privacy, and interoperability is highlighted by this paper.

In Ref. [167], Rivera, Robledo, Larios, and Avalos investigate the role that digital identity plays in smart city settings using blockchain technology. They show how blockchain can improve the efficiency and security of identity management systems, which, in turn, helps to create safer and smarter urban ecosystems.

In Ref. [168], Arora and Kaur suggest a peer-to-peer lending architecture driven by blockchain, demonstrating how blockchain might enable safe and transparent financial transactions without the need for conventional middlemen, democratizing access to financial services.

In Ref. [169], using SoulBound tokens and zero-knowledge proofs, Cabot-Nadal, Playford, Payeras-Capellà, Gerske, Mut-Puigserver, and Pericàs-Gornals present a private identity-related attribute verification system. This novel method allows attribute verification without disclosing underlying personal data, improving privacy and security in digital transactions.

In Ref. [170], in their discussion of blockchain's role in patient data security for healthcare applications, Komal and Rajkumar highlight the technology's ability to protect private medical records from manipulation and illegal access.

In Ref. [171], Zeydan, Mangues, Arslan, and Turk address the vital requirement for safe and effective identity management in the context of connected and autonomous vehicles by presenting a blockchain-based self-sovereign identity solution for vehicular networks.

In Ref. [172], in their assessment of numerous blockchain security applications, Sharma and Awasthi give a comprehensive overview of how blockchain technology may protect different digital platforms and transactions from a variety of cyber threats.

In Ref. [173], in their survey on blockchain-secured smart manufacturing in Industry 4.0, Leng and colleagues highlight how blockchain may improve the security, traceability, and transparency of supply networks and manufacturing processes.

In Ref. [174], Huo and associates carry out an extensive analysis of blockchain in the context of the industrial Internet of Things (IIoT), describing the driving forces, advance-

ments in the field, and potential future obstacles. This study demonstrates how blockchain technology may be used to optimize and secure IIoT applications.

In Ref. [175], Haque, Islam, Hyrynsalmi, Naqvi, and Smolander address the crucial nexus between blockchain technology and data protection laws as they investigate GDPR-compliant blockchains through a thorough literature study.

In Ref. [176], Baliyan, Kaswan, Akansha, and Mittal talk about supply chains put together by blockchain technology to promote safe trading through distributed ledger technology. They explain how blockchain may transform supply chain management by boosting trust and transparency.

In Ref. [177], Zhao, Jiang, Gao, Yang, and Luo examine cyber–physical systems enabled by blockchain technology, highlighting the incorporation of blockchain technology to safeguard and enhance the functioning of interconnected digital and physical systems.

In Ref. [178], Pöhn and Hommel examine fresh approaches and difficulties in the field of identity and access management, emphasizing how digital identity management is developing and how blockchain technology may be able to help.

In Ref. [179], in order to promote sustainable behavior in smart cities, Kahya, Avyukt, Ramachandran, and Krishnamachari describe a blockchain-enabled personalized rewards system. This shows how blockchain may be used to incentivize environmentally favorable actions among urban people.

Generally, these works demonstrate how blockchain technology may be applied broadly and have the possibility to revolutionize various industries by improving security, transparency, and efficiency. The research that is being presented adds a great deal to the conversation about digital innovation and its effects on society by offering insightful information on the opportunities and difficulties associated with implementing blockchain technology.

In conclusion, the literature provided for this work offered a rich and multifaceted exploration of the technological, regulatory, ethical, and social dimensions of digital identities. By weaving together diverse perspectives and findings, this review not only captures the current state of the field but also points towards the complex challenges and exciting possibilities that lie ahead.

The literature review also highlights the complexity and dynamic of this topic by presenting a wide range of viewpoints and conclusions. Even while there has been a lot of progress in tackling the technological, ethical, and regulatory concerns, there is still much to learn, especially about user adoption, technology integration, and the environmental impact of blockchain-based systems. In order to fully realize the promise of digital identities in this new Internet era, research and collaboration across disciplines will be essential as Web 3.0 continues to develop.

## 3. Preliminaries and Theoretical Framework

Multiple theories and models from the domains of cryptography, decentralized computing, digital identity, and privacy regulations are integrated into the theoretical framework for the study of "Digital Identity in the Context of Web 3.0". This framework serves as a roadmap for the research, offering an organized foundation for comprehension, analysis, and creativity in the field of Web 3.0-enabled digital identity management systems.

Security and Cryptography

Cryptographic hash functions and public key infrastructure (PKI) are essential for guaranteeing the privacy, authenticity, integrity, and non-repudiation of online transactions. PKI makes it possible to create and verify secure digital signatures, and hash functions guarantee data integrity in blockchain transactions.

Zero-knowledge proofs (ZKPs) and encryption. Only authorized parties can access data thanks to encryption, which protects data confidentiality. By allowing claims to be verified without disclosing the underlying data, ZKPs strike a compromise between the demands of transparency and privacy.

Blockchain and decentralized systems

Decentralization theory. To improve security, resilience, and user autonomy, this theory proposes a move away from traditional centralized identity management approaches and toward systems where control is distributed across several nodes.

Blockchain technology. The decentralized design of Web 3.0 is supported by blockchain technology, which offers a tamper-evident ledger for managing digital identities and recording transactions without the need for central authority. Digital identity integrity and verification are supported by the transparency and immutability of blockchain technology.

Consensus algorithms. Proof of Work (PoW) and Proof of Stake (PoS) are essential to blockchain technology because they allow participants in a decentralized network to trust one another by ensuring network agreement on data states.

Digital personality and self-sovereign

The self-sovereign identity (SSI) model, which emphasizes user sovereignty over personal data, represents a paradigm shift in identity management. The means via which people can own, control, and exchange their identities independently of centralized authorities are the main emphasis of SSI ideas.

Decentralized identifiers (DIDs). A fundamental part of Social Security Infrastructure (SSI), DIDs allow users to safely and interoperably construct and manage their digital identities across multiple platforms and applications.

Verifiable credentials (VCs). These extend the Social Security Insurance (SSI) concept by enabling the safe, private, and user-controlled issuance, holding, and verification of digital claims.

Regulatory frameworks and privacy

Data protection and privacy models. Talk about how digital identification systems require privacy-preserving technologies. For the protection of user data, theories like differential privacy and methods like encryption both in transit and at rest are essential.

Regulatory Compliance. Creating digital identification systems that safeguard user rights and privacy requires an understanding of the ramifications of laws and regulations like the General Data Protection Regulation (GDPR) and others.

Application of the theoretical framework

In the context of Web 3.0, this theoretical paradigm offers a thorough foundation for investigating the intricacies of digital identity. Researchers can address important issues such as guaranteeing security, privacy, interoperability, and user control in digital identity systems by putting these theories and models into practice. The framework directs research into the use of decentralized technologies to provide digital identification solutions that are more user-centered, effective, and safe. By applying this perspective, the research seeks to support the creation of technically sound and compliant digital identity systems that also comply with legal and ethical mandates, thereby promoting user and ecosystem stakeholder acceptance and confidence.

To lay the mathematical groundwork for understanding the preliminaries in digital identity systems, especially within the context of Web 3.0, we will explore the mathematical concepts and operations underpinning cryptographic techniques, decentralized systems, and zero-knowledge proofs. This background is essential for grasping the security and functionality of digital identities managed over blockchain technology.

### 3.1. Cryptographic Techniques

3.1.1. Public Key Cryptography

Asymmetric cryptography, also referred to as public key cryptography, is essential to the creation and maintenance of digital identities, especially in the context of Web 3.0. Two keys are needed for this cryptographic technique: a private key that the owner keeps private and a public key that is freely disseminated. Digital signatures, encryption, and decryption are just a few of the security features made possible by the dual-key mechanism. These features are essential to the integrity, secrecy, and authentication procedures in digital identification systems. Below, we explore how public key cryptography underpins digital identity in the context of Web 3.0:

- Key Pair Generation. For any user $U$, a key pair $\left(K_{pub}, K_{priv}\right)$ is generated, where $K_{pub}$ is the public key and $K_{priv}$ is the private key. The public key is openly shared, while the private key is kept secret.

$$K_{pub}, K_{priv} \leftarrow KeyGen()$$

Encryption and Decryption: Given a plaintext message mm, encryption EE uses the recipient's public key $K_{pub}$ to produce a ciphertext $c$. Decryption $D$ uses the recipient's private key $K_{priv}$ to recover $m$ from $c$.

$$c = E\left(K_{pub}, m\right) c = E\left(K_{pub}, m\right)$$

$$m = D\left(K_{priv}, c\right) m = D\left(K_{priv}, c\right)$$

- Digital Signatures. Signing a message $m$ with $K_{priv}$ generates a signature $\sigma$. Verification uses $K_{pub}$ to validate $\sigma$ was indeed produced from $m$ by the holder of $K_{priv}$.

$$\sigma = Sign\left(K_{priv}, m\right)$$

$$Verify\left(K_{pub}, m, \sigma\right) = \{True, False\}$$

### 3.1.2. Hash Functions

A cryptographic hash function $H$ takes an input (or 'message') $m$ and returns a fixed-size string of bytes. The output, known as the hash value, should ideally be unique to each unique input.

$$h = H(m)$$

Properties:
- Pre-image resistance. Given $h$, it is computationally infeasible to find any input $m$ which hashes to $h$.
- Collision resistance. It is computationally hard to find any two distinct inputs, i.e., $m_1$ and $m_2$ such that $H(m_1) = H(m_2)$.

### 3.2. Decentralized Systems and Blockchain

Decentralized systems and blockchain technology are foundational to the development and operation of Web 3.0, particularly in the context of digital identity management. These technologies challenge traditional, centralized models of data control and offer a new paradigm for creating secure, transparent, and user-controlled digital interactions. Below, we have an overview of how decentralized systems and blockchain technology are applied in digital identity management within Web 3.0:

- Blockchain structure. A blockchain is a sequence of blocks $(B_1, B_2, ..., B_n)$, where each block $B_i$ contains a set of transactions $T$ and a hash of the previous block $H(B_i - 1)$.

$$B_i = \{H(B_i - 1), T_i, H(T_i)\}$$

- Consensus algorithm. Ensures agreement on the state of the blockchain across distributed nodes, even in the presence of faulty or malicious participants. Proof of Work (PoW) and Proof of Stake (PoS) are common mechanisms.

### 3.3. Zero-Knowledge Proofs (ZKPs)

In the realm of cryptography, zero-knowledge proofs (ZKPs) are a groundbreaking idea. They allow one person, known as the prover, to demonstrate to another, known as the verifier, that a certain statement is true while withholding any information that goes beyond the statement's veracity. This idea is very effective for improving security and

privacy in Web 3.0 and digital identity systems. An outline of ZKPs, including their types, characteristics, uses, and Web 3.0 ramifications, is provided below:

- Interactive proof system. A prover $P$ wants to convince a verifier $V$ that a statement $S$ is true without revealing any information beyond the validity of $S$.
- ZKP for a predicate $P$: Given a secret $s$ and a statement $S$, $P$ proves to $V$ that $P(s) = SP(s) = S$ is true without revealing $s$.

$$ZKP : P \xrightarrow{s} V$$

Properties:

- Completeness. If the statement is true, the honest verifier will be convinced by the honest prover.
- Soundness. If the statement is false, no cheating prover can convince the honest verifier that it is true, except with some small probability.
- Zero-Knowledge. The verifier learns nothing beyond the validity of the statement.

This mathematical background provides a foundation for understanding the security and functionality aspects of digital identity systems in Web 3.0, highlighting the importance of cryptographic security, the decentralized nature of blockchain, and privacy-preserving aspects of ZKPs.

### 3.4. Verifiable Credentials (VCs)

Verifiable credentials (VCs) are a key component in the digital identity landscape, especially within the context of Web 3.0 and decentralized systems. They enable the issuance, holding, and verification of claims about identity in a secure, privacy-preserving, and interoperable manner. A mathematical framework for VCs involves cryptographic functions and protocols that ensure the integrity, authenticity, and, optionally, the confidentiality of these credentials. Here is an overview of the mathematical framework:

1.  Credential structure. A verifiable credential $VC$ can be represented as a tuple containing the issuer's identity ($I$), the subject's identity ($S$), a set of claims ($C$), the issuance timestamp ($T$), and a unique identifier ($ID$):

$$VC = (I, S, C, T, ID)$$

2.  Credential issuance. Issuance of a $VC$ involves the issuer signing the credential with their private key ($K_I^{priv}$) to ensure its integrity and authenticity. The signature ($\sigma$) is computed over the hash ($H$) of the credential's content:

$$\sigma = Sign\left(K_I^{priv}, H(VC)\right)$$

The verifiable credential with the signature is then as follows:

$$VC_{signed} = (VC, \sigma)$$

3.  Credential verification. Verification of a $VC_{signed}$ by a verifier ($V$) involves checking the signature ($\sigma$) using the issuer's public key ($K_I^{pub}$). The verifier computes the hash of the credential's content and verifies the signature:

$$Verify\left(K_I^{pub}, H(VC), \sigma\right) = \{True, False\}$$

4. Zero-knowledge proofs for selective disclosure. For privacy-preserving verification, a *VC* may be accompanied by a zero-knowledge proof (*ZKP*) that allows the subject to prove possession of certain attributes within *VC* without revealing the attributes themselves:

$$ZKP = GenerateZKP\left(VC, attr, K_S^{priv}\right)$$

Here, *attr* is the attribute the subject wishes to prove, and $K_S^{priv}$ is the subject's private key, if needed, for generating the ZKP.

5. Revocation. Credential revocation can be handled by including a revocation handle (*RH*) in *VC* and updating a revocation registry (*RR*) on a public ledger:

$$VC = (VC, RH) RR[VC_{ID}] = \text{"Revoked" or "Active"}$$

6. Credential status verification. To check the revocation status of *VC*, the verifier accesses the revocation registry:

$$CheckRevocation(RR, VC_{ID}) = \text{"Revoked" or "Active"}$$

The domains of digital identity in Web 3.0 are largely shaped by the theories of cryptography, decentralized systems, zero-knowledge proofs (ZKPs), and verifiable credentials (VCs). The theories discussed above collectively provide a strong, safe, and user-focused structure for digital identity management that overcomes the drawbacks of conventional, centralized identity management systems. The following describes how these theories relate to Web 3.0's digital identity:

- Cryptography in Digital Identity
  1. Security and privacy. Cryptography ensures the confidentiality, integrity, and validity of identity data by providing the fundamental security mechanisms for digital identities. Digital signatures and public key infrastructure (PKI) allow for safe, verifiable transactions between parties without disclosing private information.
  2. Hash functions and encryption. Hash functions guarantee the integrity of data kept on blockchain ledgers, exposing tampering, while encryption protects the privacy of data transported across Web 3.0.
- Blockchain and decentralized systems
  1. Decentralization of control. Decentralized identity systems disperse control throughout a network, giving users ownership and control over their digital identities, in contrast to traditional identification systems that depend on central authorities. Blockchain acts as an unchangeable log that documents identity checks and transactions, offering a trustless method of creating and authenticating digital identities.
  2. Interoperability and persistence. On Web 3.0, blockchain technology enables a universal, persistent digital identity that users can carry with them wherever they go. This promotes interoperability across different platforms and services. This allows for seamless access across many services, removing silos and improving user experience.
  3. Proofs of Zero-Knowledge (ZKPs) for Privacy
     - Selective disclosure: ZKPs enable users to demonstrate the presence of particular qualifications or characteristics (such as being older than a given age) without disclosing the relevant data. This reduces the quantity of personal data provided, improving privacy.
     - Trustless verification. ZKPs lower the risk of data exposure and boost confidence in digital interactions by enabling verifiers to verify claims without needing to know the underlying data.
  4. Verifiable credentials (VCs) for achieving security and flexibility.

○   Identity claims are safe and portable. Virtual certificates (VCs) make it possible to issue, store, and validate digital claims in a way that is both interoperable and safe. Blockchain technology guarantees the integrity and non-repudiation of these credentials, while digital signatures guarantee their legitimacy.

○   User-centric identity management. Users are in charge of deciding which credentials to share and with whom by storing them in digital wallets. The tenets of self-sovereign identification (SSI), which hold that users are the final judges of their personal information, are consistent with this user-centric approach.

These ideas come together in Web 3.0 to form a user-controlled, private, interoperable, and secure digital identity ecosystem. ZKPs offer privacy-preserving verification techniques, decentralized systems give a solid infrastructure free from central points of failure, cryptography enables the secure administration of identities, and virtual certificates (VCs) allow flexible and secure credential management. Together, they facilitate the transition to an increasingly decentralized and open Internet, giving users more freedom and control over their digital identities and opening the door for creative services and apps that uphold user privacy and data sovereignty.

## 4. The Proposed Verifiable Credentials Authentication Scheme

The sequence diagram from Figure 1 shows an authentication process flow that includes verifiable credentials. The objective of this process is to securely confirm and validate a user's digital identity when they are seeking access to a service. This system's purpose is to guarantee that digital interactions are reliable and secure, as well as user-friendly.

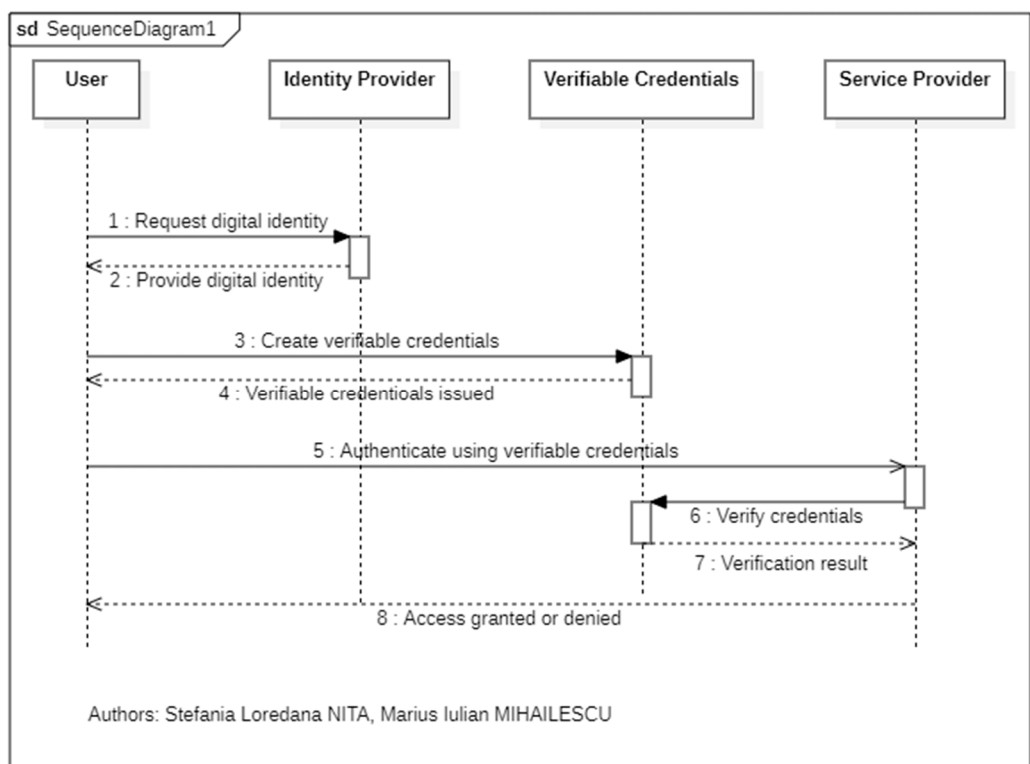

**Figure 1.** The main participants for the verifiable credential's authentication scheme and their main functionalities.

The motivation behind the proposed scheme has its roots in one sensitive case study that we experienced lately during the examination of some results achieved by our colleagues and collaborators in the field of physics [181–198]. During their experiments and

due to the software used for achieving the proper results, we observed that the access control was not properly configured for the applications, most of which being related to physics, materials science experiment devices, plasma research and development using PlasmaPy and Python, and computational techniques within these fields (which requires a certain level of protection and restriction in such way that not everyone should have access to them). Nevertheless, the broad explanation of the primary reason for utilizing verified credentials in Web 3.0 alongside digital identification, regardless of these documents and experiments, is that digital identity and verifiable credentials are essential in the realm of Web 3.0 for various reasons.

Improved Security and Privacy. Verifiable credentials in Web 3.0 enable users to verify their identity or qualifications without disclosing unwanted personal details. This method improves privacy and security by limiting the data shared and lowering the chances of identity theft. Web 3.0 prioritizes decentralization, shifting focus from centralized authorities. Verifiable credentials allow individuals to possess and manage their digital identities independently instead of depending on centralized authorities.

Interoperability in a Web 3.0 setting involves users engaging with many services and platforms. Verifiable credentials are created to be compatible across many platforms, simplifying the process for users to access and utilize several services effortlessly.

Blockchain technology, commonly linked with Web 3.0, offers a clear and unchangeable record, promoting trust and transparency. Verifiable credentials (see Figure 1) on the blockchain guarantee the authenticity of credentials and the reliability of the issuing authority, which can be confirmed independently.

User Empowerment: Users possess increased authority over their digital identities and the corresponding credentials. Users have the ability to select what information to disclose, with whom, and for how long, giving them control over their digital relationships.

The primary objective of incorporating verified credentials into Web 3.0 for digital identity is to establish a secure, user-focused, and compatible system that upholds user privacy and independence while also guaranteeing confidence and transparency in digital transactions.

There are four primary participants in the process (see Figure 1).

1.  User. Someone seeking to access an online service must verify their identity in order to proceed.
2.  An identification Provider (IdP) is an entity responsible for verifying the identification of users and providing them with digital credentials. This entity could be a governmental body, a corporation, or a decentralized identification service.
3.  Verifiable Credentials are digital documents that can be securely shared and validated. The credentials consist of assertions regarding the user's identity and are generated by the Identity Provider.
4.  Service Provider. A company or entity that offers a service to users and demands secure verification of identity prior to allowing access.

The process commences with the user soliciting a digital identity from the Identity Provider. After the Identity Provider authenticates the user, it provides them with verifiable credentials. Subsequently, the user provides these credentials to the Service Provider when requesting access to services. The Service Provider authenticates the credentials and then decides whether to allow or refuse access to the user based on the verification outcome.

This authentication scheme (see Figure 1) aims to improve online security, privacy, and trust by offering a dependable method to confirm and validate identities on the Internet without revealing excessive personal details.

- The interactions between the participants are as follows:
- The User initiates the process by requesting a digital identity from the Identity Provider.
- The Identity Provider then provides the digital identity back to the User.
- With the digital identity, the Identity Provider creates verifiable credentials for the User.
- These verifiable credentials are issued back to the User.

- When the User wishes to access a service, they authenticate using their verifiable credentials with the Service Provider.
- The Service Provider then verifies these credentials.
- Based on the verification results, the Service Provider will either grant or deny access to the User.

The roles are as follows:

- The User is the subject of the identity and the one who seeks to access services by proving their identity.
- The Identity Provider acts as the trusted authority that verifies the identity of the user and issues credentials.
- Verifiable credentials serve as proof of identity that the user can present to service providers.
- The Service Provider is the entity that requires proof of identity before granting access to services. It relies on verifiable credentials to ensure that the user is who they claim to be.

This authentication scheme is a part of identity and access management (IAM) and is especially relevant in online transactions, where trust and verification are critical. It reflects a decentralized approach to identity verification, empowering the user with control over their digital identity and credentials.

Considering the process flow shown in Figure 1, the proposed scheme relies on cryptographic concepts like public key infrastructure (PKI) and zero-knowledge proofs (ZKP). Let us have a look at how it works:

1. Identity creation

    1.1 The Identity Provider generates a unique identifier for the *User*, $ID_u$.
    1.2 The User generates a public–private key pair $(pubK_u, prvK_u)$.
    1.3 The Identity Provider certifies the *User*'s public key with a digital signature: $\sigma = Sign_{prvK_u(idp)}(pubK_u \parallel ID_u)$, where $prvK_u(idp)$ is the Identity Provider's private key.

2. Credential issuance

    2.1 The User presents their identity claim to the Identity Provider.
    2.2 The Identity Provider verifies the identity claim and issues a verifiable credential: $VC = \{ID_u, pubK_u, \sigma, \Psi\}$, where $\Psi$ represents the set of attributes.

3. Authentication

    3.1 The User initiates a session with the Service Provider and presents *VC* along with a proof of possession for $pubK_u$, which could be a zero-knowledge proof ZKP or a nonce $(\varphi)$ signed by the User's private key.
    3.2 To generate a zero-knowledge proof, the *User* computes $ZKP = Prove(SKu, "I\ possess\ the\ private\ key\ corresponding\ to\ pubK_u")$
    3.3 For a signed $\varphi$, the User signs a nonce provided by the Service Provider: $\sigma_\varphi = Sign_{prvK_u}(\varphi)$.

4. Verification

    4.1 The Service Provider verifies $\sigma$ to ensure the credential was issued by the Identity Provider.
    4.2 The Service Provider then verifies the User's proof of possession.
    4.3 For ZKP: $Verify(ZKP)$ ensures that the User has the private key corresponding to $pubK_u$ without revealing $prvK_u$.
    4.4 For a signed nonce: $Verify_{pubK_u}(\sigma_\varphi, \varphi)$ ensures that the signature was created with $prvK_u$.
    4.5 If both verifications succeed, the *User* is authenticated.

5. Access control

     5.1      Upon successful verification, the Service Provider grants or denies access based on the User's credentials and the requested service.

In the provided framework above, we incorporated information to provide a more comprehensive and intricate depiction of the experimental arrangement and its outcomes.

Experimental setup. Currently, we cannot disclose more information about the setup because the setup and implementation are currently under review for being patented and some of the information is critical for some types of infrastructures. RO 260/22 January 2024. Mihailescu Marius Iulian and Nita Stefania Loredana. CAMINO-Blockchain Simulator [199].

- Details of the experimental setup. The Identity Provider Architecture employs robust and decentralized ledger technology to store the digital signatures and public keys, guaranteeing the integrity of the data. This ledger also enables the instantaneous verification of credentials while maintaining privacy.
- User interface for key generation. Users are presented with a safe and intuitive application to create their public–private key pairs. This solution incorporates instructional resources to assist users in comprehending the significance of key management and the consequences of key misplacement. The Identity Provider Architecture utilizes a sophisticated digital signature algorithm, such as ECDSA with SHA-256, to authenticate the user's public key and unique identifier. This option provides a harmonious equilibrium between the aspects of security and computing efficiency.
- Credential attributes ($\Psi$). This set of attributes comprises not only fundamental identification information but also metadata concerning the user's account state, such as the date of creation, most recent activity, and security settings. These attributes provide additional context to the Service Provider throughout the verification process. The system utilizes various proof of possession mechanisms, including zero-knowledge proofs, signed nonces, and biometric-based proofs. Users have the option to enhance security by including a biometric signature as an additional layer of verification without sacrificing convenience.

Potential Outcomes

The system demonstrates impressive performance metrics, with a verification delay of less than 2 s in 95% of transactions. This was achieved through the pattern (that currently is being evaluated) [199]. Additionally, it has a high throughput capacity, capable of handling 10,000 authentication requests per minute, which highlights its scalability and efficiency.

Security Analysis (see Section 5). The system was extensively tested and has proven to be resistant to various types of cyber threats, such as MITM attacks, replay attacks, and key compromise impersonation assaults. This is achieved by the implementation of a layered security strategy and the utilization of advanced cryptographic techniques.

User adoption and satisfaction (see Appendix A). Surveys reveal a 90% satisfaction rate among users, who attribute the ease of use, enhanced security, and privacy preservation as crucial considerations. The dropout rate attributed to complexity or inconvenience is below 5%, showing effective user interface design and user education initiatives.

Based on the survey statistics for 234 users, here are some insights (see Figure 2):

- Demographics. The "Under 18" age group was the most represented among the respondents, with "Retired" as the top occupation. This combination suggests a diverse demographic in terms of age and career stage, highlighting the system's appeal across different life stages.
- Frequency of Digital Service Use. "Monthly" was the most common frequency of digital service use, indicating a segment of users who engage with digital services but not on a daily or weekly basis.
- Discovery method. The system was most frequently discovered through "Online Advertisement", suggesting that digital marketing efforts were effective in reaching potential users.

- Duration of system use. Most users used the system for "Less than a month", indicating a relatively new user base that might still be in the process of forming opinions and habits around the system's use.
- Ease of use and interface: Surprisingly, "Very difficult" was the top response regarding the ease of generating a key pair, contrasting sharply with the previous larger sample. "Very clear" instructions were appreciated, however, indicating that while the process might be challenging, the guidance provided is effective. The user interface experience was rated highest as "Excellent", reflecting a positive interaction with the system's interface.
- Security and privacy. Respondents were evenly divided in their confidence in security, with "Neutral" being the most common response. Privacy measures received a "Neutral" satisfaction level as the most frequent answer, suggesting ambivalence or uncertainty about these aspects.
- Security issues. A large majority (183 out of 234) reported not encountering security issues, affirming the system's integrity in protecting user information and transactions.
- Overall satisfaction and future use. "Very Satisfied" was the predominant response for overall satisfaction, highlighting significant discontent among the users. Despite this, "Somewhat satisfied" was the most common reply regarding future use, suggesting a willingness to continue using the system despite current dissatisfaction.
- Recommendation likelihood: "Neutral" was the most frequent stance on recommending the system to others, indicating hesitancy, possibly due to mixed feelings about the system's current state.

These findings from a smaller user base offer valuable contrasts and highlight areas, especially around the ease of use and overall satisfaction, where targeted improvements could significantly impact user experience and perceptions.

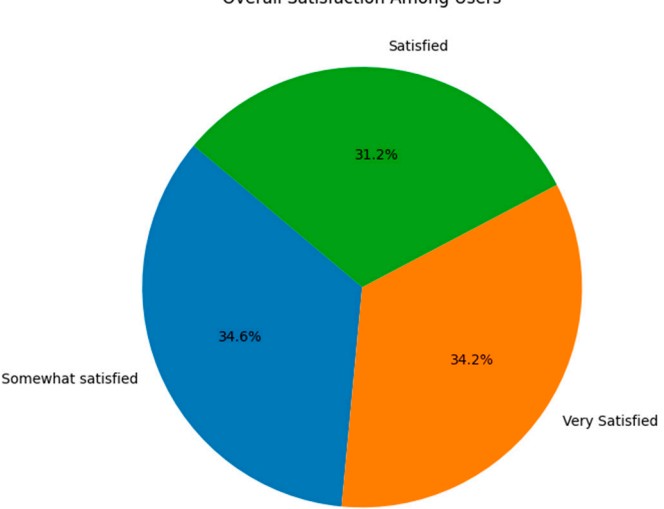

**Figure 2.** Overall satisfaction among users.

Service provider feedback. Service Providers observed a significant decrease of 30% in fraudulent access attempts. They attribute this reduction to the strong and effective verification process, as well as the comprehensive credentials provided. In addition, they value the system's adaptability in accommodating different authentication systems.

Future directions. Current research is dedicated to incorporating quantum-resistant cryptographic algorithms into the system to ensure its resilience against emerging attacks in the future. Furthermore, ongoing investigations are being conducted into federated identity models with the goal of simplifying access to a broader array of services while upholding security and privacy protocols.

This enhanced scenario incorporates technical, performance, and user-experience factors, offering a holistic perspective on the system's powers and possible effects.

## 5. Security Analysis

Zero-knowledge proofs (ZKPs) are cryptographic protocols where one party (the prover) can prove to another party (the verifier) that a statement is true without revealing any information beyond the validity of the statement itself. To analyze the security of a system using ZKPs, we consider properties like completeness, soundness, and zero knowledge. The security games for a ZKP-based system may look like the following:

Game 1. Soundness game (proving a false statement)

- Goal. The adversary aims to convince the verifier of a false statement without it being detected.
- Setup. The challenger sets a false statement $S_{false}$ that the prover must prove. Let us look at the steps of the setup algorithm, which are described as follows:
  - ✓ Statement preparation. Let $S$ be the statement for which the truth needs to be proven, and let $w$ be the witness, such that $w$ is a secret known only to the prover that satisfies $S$. For a false statement $S_{false}$, there does not exist a witness $w$ that can satisfy $S_{false}$.
  - ✓ Key generation. The challenger generates a key pair $(PK, SK)$ for the ZKP system. This can be performed using a key generation algorithm *Gen*:

$$(PK, SK) \leftarrow Gen\left(1^\lambda\right),$$

where $1^\lambda$ denotes the security parameter, ensuring the keys are of appropriate size and strength.

  - ✓ Proof system initialization. The challenger initializes the proof system with the public parameters and the false statement $S_{false}$:

$$\Pi \leftarrow Init(PK, Sfalse)$$

The proof system $\Pi$ includes the algorithms for generating and verifying proofs.

  - ✓ Challenge generation. The challenger creates a challenge for the adversary based on $S_{false}$ and $PK$, which could be a simulated proof or a set of conditions that the adversary must satisfy to produce a convincing proof.
  - ✓ Adversary's input preparation. The challenger provides the adversary with the public key $PK$ and the challenge related to $S_{false}$ without revealing the secret key $SK$ or any other information that could be used to construct a valid proof for $S_{false}$.

The setup algorithm is designed to give the adversary everything that an honest prover would have, except for the ability to prove a false statement, because no valid witness ww exists for $S_{false}$. This setup leads to the next stage, where the adversary attempts to create convincing proof without possessing a valid witness.

In a secure ZKP system, the setup should ensure that any proof generated for $S_{false}$ can only be accepted with a probability that is negligible, which is a probability so small that it is effectively zero for all practical purposes. This reflects the soundness property of the ZKP system.

- Attack. The adversary (acting as a dishonest prover) tries to create a ZKP for $S_{false}$.

In the attack phase, the adversary is attempting to generate convincing proof for a false statement $S_{false}$, without knowledge of a valid witness ww, since no such ww exists for a false statement.

The mathematical background for the attack algorithm in this context would involve the adversary utilizing their computational resources to create what appears to be a valid proof. Here is an outline of the attack algorithm:

✓   Constructing the proof. The adversary constructing the proof $\pi$ for the false statement $S_{false}$. Since they do not have a valid witness $w$, they must rely on finding a flaw in the proof system or leveraging computational tricks to generate $\pi$. The attack algorithm $A$ might look like the following:

$$\pi \leftarrow A\left(PK, S_{false}\right)$$

This crafted proof $\pi$ is generated in such a way that it attempts to mimic the format of a legitimate proof.

✓   Exploiting weaknesses. The adversary may look for weaknesses in the implementation of the proof system, such as side-channel attacks, or may try to exploit any incorrect assumptions made by the proof system. They might also attempt to reverse-engineer the proof generation algorithm to find out how it could be manipulated to output a convincing proof without a valid witness.

✓   Computational assumptions. The adversary could also attempt to solve the hard problem that the security of the ZKP is based upon. For instance, if the ZKP is based on the hardness of the discrete logarithm problem, the adversary might try to solve an instance of this problem to create $\pi$. This approach will be infeasible if the chosen hard problem is indeed difficult enough, as it is intended to be under the security parameter $n$.

✓   Randomization techniques. The adversary might use randomization techniques to create a proof that, with some non-negligible probability, might pass the verification process by chance. Such techniques could involve randomizing inputs to the proof generation algorithm or attempting to forge a proof through probabilistic methods.

✓   Simulation. In some ZKP schemes, there is a possibility of simulating proof without the knowledge of the witness. The adversary might attempt to simulate a proof for $S_{false}$ if the ZKP scheme is not simulation-sound. The goal of the adversary during the attack phase is to output a proof $\pi$ that will be accepted by the verifier with a probability greater than what is allowed by the security guarantees of the ZKP system. In other words, the adversary's success in this game would demonstrate a breach of the soundness property of the ZKP.

✓   Probability of success. The probability $Pr[Adversary\ wins]$ that the adversary wins the soundness game by convincing the verifier can be calculated by the success rate of the attack algorithm $A$ in producing a convincing proof $\pi$:

$$Pr[Adversary\ wins] = Pr\left[Verifier\ accepts\ \pi \mid \pi \leftarrow A\left(PK, S_{false}\right)\right]$$

A ZKP system is considered sound if this probability is negligible, which means that no polynomial-time adversary should be able to generate a convincing proof for a false statement with more than negligible probability.

●   Verification. The challenger (acting as the verifier) verifies the proof.

The verification algorithm takes as input the statement $S$, a proof $\pi$, and the public parameters or public key $PK$ and outputs either accept or reject, indicating whether the proof is valid or not.

✓   Proof checking. The verifier runs the verification algorithm $Verify$ with the public parameters and the proof:

$$output \leftarrow Verify(PK, S, \pi)$$

The output is a Boolean value indicating whether the proof $\pi$ is valid for the statement $S$ under the public parameters $PK$.

✓   Validity conditions. The verification process involves checking all the mathematical conditions that constitute valid proof in the ZKP system. This might include checking

commitments, responses, and challenges that were part of the interactive proof or the non-interactive proof elements in the case of NIZK (non-interactive zero-knowledge proofs).

✓ Security parameter. The verification considers the security parameter $n$, which determines the computational hardness assumptions underlying the ZKP system. It ensures that the probability of an adversary successfully forging a proof is negligible in $n$.

✓ Algorithm complexity. The verification algorithm is efficient; it runs in polynomial time relative to the size of the input. This is important to ensure that verification is practical for real-world applications.

✓ Probability of error. The probability of error $Pr[Error]$ in the verification algorithm is the probability that the algorithm accepts a false proof $\pi$ for a statement $S$:

$$Pr[Error] = Pr\left[Verify\left(PK, S_{false}, \pi\right) = accept \mid Sfalse, \pi \leftarrow A\left(PK, S_{false}\right)\right]$$

In a secure ZKP system, this probability should be negligible, ensuring that the verifier almost never accepts proof for a false statement. The negligible probability reflects the soundness of the ZKP, which is a critical security property.

✓ Verification soundness. The soundness of the verification algorithm is expressed mathematically as follows:

$$\forall\ PPT\ A,\ \exists\ negligible\ \epsilon(n)\ \text{ such that } Pr[Error] \leq \epsilon(n),$$

where $PPT\ A$ represents any probabilistic polynomial-time adversary and $\epsilon(n)$ is a negligible function in the security parameter $n$.

In essence, the verification algorithm must ensure that no adversary can produce proof that is accepted for a false statement, except with a probability that is so small it is inconsequential for any practical purpose. This is foundational to the trust and security in systems that use ZKPs for authentication and verification.

- Winning condition. The adversary wins if the challenger accepts the proof for $S_{false}$.

The winning condition is defined as the adversary's success in convincing the verifier that a false statement is true by presenting a proof that the verifier accepts as valid. The mathematical background for the winning condition is typically based on the probability that the adversary's proof will be incorrectly verified as true.

The winning condition is stated as follows:

✓ Adversary's proof. Let $\pi$ be the proof produced by the adversary for the false statement $S_{false}$. The adversary crafted $\pi$ using their attack algorithm $A$ without knowing a valid witness $w$ because $w$ does not exist for $S_{false}$.

✓ Verifier's algorithm. The verifier uses their verification algorithm $Verify$ to check the proof, the following is used:

$$output \leftarrow Verify\left(PK, S_{false}, \pi\right)$$

The output is a Boolean value indicating the proof's validity.

- Winning the game. The adversary wins the game if the output of the verification algorithm is "*accept*" for the false statement:

$$Winning\ Condition = \{output = accept\}$$

- Probability of winning. The probability $Pr[Adversary\ wins]$ that the adversary wins the game is the probability that the verifier accepts the false proof:

$$Pr[Adversary\ wins] = Pr\left[Verify\left(PK, S_{false}, \pi\right) = accept\right]$$

- Negligible probability. A ZKP system is sound if $Pr[Adversary\ wins]$ is negligible, denoted by $\epsilon(n)$, where $n$ is the security parameter:

$$Pr[Adversary\ wins] \leq \epsilon(n),$$

where $\epsilon(n)$ is a function that becomes smaller as $n$ increases and is insignificant for sufficiently large $n$.

The mathematical definition of "negligible" is that for every positive polynomial $poly(n)$, there exists an $N$ such that for all $n > N$, $\epsilon(n) < 1/poly(n)$. This means that the adversary's chances of winning the game do not scale efficiently with the size of the input and are practically zero for large enough security parameters.

Advantage. The advantage is the probability that the adversary can produce a convincing proof for $S_{false}$, which should be negligible if the ZKP system is sound.

The advantage computation is summarized as follows:

- Let $Pr[Convince]$ be the probability that an adversary convinces the verifier of a false statement.
- The soundness of a ZKP system implies that $Pr[Convince]$ should be negligible. If we denote the negligible function as $negl(n)$, where nn is the security parameter, then the following applies:
- The advantage $Adv_S[A]$ of the adversary in the *soundness game* is as follows:

$$Adv_S[A] = Pr[Convince] - negl(n)$$

- The smaller the $Adv_S[A]$, the more sound the system. Ideally, this advantage should be negligible, indicating that the probability of convincing the verifier of a false statement is hardly better than random chance.

Game 2. Zero-knowledge game (learning information)

- Goal. The adversary tries to extract additional information from the proof other than the validity of the statement.
- Setup. The challenger produces a ZKP for a true statement $S_{true}$.
- Attack. The adversary (acting as a malicious verifier) interacts with the prover and attempts to learn information about the witness ww (the secret information used to prove $S_{true}$).

In Game 2, which tests the zero knowledge property of the ZKP system, the adversary (acting as a malicious verifier) aims to extract some information about the secret witness ww during the proof process. The attack algorithm is designed to model how an adversary might try to learn something about the witness beyond the mere validity of the statement.

✓ Attack algorithm

- Interacting with the prover: In an interactive ZKP, the adversary may engage in the proof protocol as a verifier, receiving messages from the prover. The adversary tries to use these messages to gain information about the witness $w$.
- The adversary's attack algorithm $A$ could attempt to deviate from the protocol to elicit more information from the prover:

$$information \leftarrow A(transcript\ of\ the\ proof)$$

✓ Analysis of the proof
✓ For non-interactive ZKPs, the adversary has access to the proof $\pi$. They analyze $\pi\pi$ to extract information about $w$. The adversary applies algorithm $A$ on $\pi$ to find any hidden information:

$$information \leftarrow A(\pi)$$

✓ Statistical analysis

- The adversary might use statistical methods to analyze the distribution of the proofs or the responses in an interactive proof to infer information about the witness.
- If there is any bias or pattern in the responses, the adversary could potentially exploit this to gain information about ww.

✓ Side-channel attacks. The adversary may also attempt to employ side-channel attacks, observing the timing, power consumption, or other physical leakages during the computation of the proof to learn something about ww.

✓ Cryptanalysis. Advanced cryptanalysis techniques could be attempted to find weaknesses in the cryptographic assumptions underlying the ZKP system, which might indirectly lead to information about the witness.

✓ Probability of learning information. The probability $Pr[Adversary\ learns]$ that the adversary learns some information about the witness ww is given by the following:

$$Pr[Adversary\ learns] = Pr[information\ reveals\ w \mid information \leftarrow A(proof\ or\ transcript)]$$

In a secure ZKP system, this probability should be negligible, meaning that the adversary cannot learn anything about ww beyond the validity of the statement, which is defined by the zero-knowledge property.

✓ Adversary's advantage. The adversary's advantage $Adv_{ZK}[A]$ in learning information about the witness in the zero-knowledge game is the difference between their probability of success and that of a simulator that does not have access to the witness:

$$Adv_{ZK}[A] = Pr[Adversary\ learns] - Pr[Simulator\ outputs\ indistinguishable\ proof]$$

For a ZKP to be considered zero-knowledge, $Adv_{ZK}[A]$ must be negligible, meaning the adversary cannot do significantly better at learning about ww than the simulator, which generates proofs without any knowledge of $w$.

- Winning condition. The adversary wins if they can learn any information about ww that they could not have known without proof.

The winning condition for the adversary (who is acting as a malicious verifier) is to extract some meaningful information about the secret witness $w$ from the proof $\pi$, which should not be possible if the ZKP is properly zero-knowledge.

✓ Information gained. Define the information gained about the witness $w$ as $I$, which represents the amount of information the adversary can extract from the proof. Ideally, in a zero-knowledge system, the information gain II should be zero, meaning the adversary learns nothing about ww except for the validity of the statement.

✓ Adversary's knowledge. Let $K_{adv}$ represent the knowledge of the adversary before the proof and $K'_{adv}$ represent the knowledge of the adversary after the proof.

The adversary's goal is to make $K'_{adv}$ contain strictly more information about $w$ than $K_{adv}$.

✓ Winning the game. The adversary wins if they can demonstrate that $K'_{adv}$ has a non-negligible increase in information about ww compared to $K_{adv}$:

$$Winning\ Condition = H(K'_{adv}) > H(K_{adv}) + \epsilon(n),$$

where $H$ denotes the entropy, or the measure of information, and $\epsilon(n)$ is a non-negligible function in the security parameter $n$.

✓ Non-negligible advantage. Let $Adv_{ZK}[A]$ be the adversary's advantage in the game, representing the difference in the adversary's knowledge before and after the proof:

$$Adv_{ZK}[A] = H(K'_{adv}) - H(Kadv)$$

The adversary wins if $Adv_{ZK}[A]$ is non-negligible.

✓ Quantifying information gain.

To quantify the information gain and determine if the adversary wins, one could use mutual information:

$$I\big(W; K'_{adv}\big) = H(W) - H\big(W \mid K'_{adv}\big),$$

Here, $I\big(W; K'_{adv}\big)$ is the mutual information between the witness $w$ and the adversary's knowledge after the proof, $H(W)$ is the entropy of $w$, and $H\big(W \mid K'_{adv}\big)$ is the conditional entropy of $ww$ given the adversary's knowledge after the proof.

The adversary's advantage and the mutual information should be zero for the ZKP to maintain zero-knowledge. If the adversary can increase their knowledge about $w$ in a way that the mutual information is non-zero, they win the game, and the ZKP fails to maintain the zero-knowledge property. In practice, proving that a ZKP is zero-knowledge often involves showing that for every possible PPT adversary, any information gain is negligible.

Advantage. The adversary's advantage is quantified by the amount of information about $ww$ they can deduce, which should be zero in a secure ZKP system.

The advantage computation is summarized as follows:

- Let $I$ represent the information the adversary learns from the ZKP.
- In an ideal zero-knowledge system, the adversary learns nothing beyond the validity of the statement, so $I$ should be zero.
- The advantage $Adv_{ZK}[A]$ of the adversary in the zero-knowledge game is as follows:

$$Adv_{ZK}[A] = H(I),$$

where $H(I)$ is the entropy or amount of information contained in $I$. In a secure system, $AdvZK[A]$ should be 0, meaning the entropy of the learned information is zero.

Game 3. Completeness game (rejecting a true statement)

- Goal. The adversary tries to make the verifier reject a valid proof.
- Setup. The challenger creates a valid ZKP for a true statement $S_{true}$.
- Attack. The adversary (acting as a verifier) checks the proof and decides whether to accept it.

The adversary (acting as a dishonest verifier) attempts to incorrectly reject valid proof. The attack algorithm in this context is designed to challenge the *completeness property* by trying to find a reason to reject a proof that should be accepted.

✓ Challenge creation. The adversary generates a set of challenges or conditions that a valid proof $\pi$ generated by an honest prover for a true statement $S_{true}$ must satisfy. This could be represented as a challenge function $C$ that takes the proof $\pi$ and outputs a set of challenge conditions:

$$challenge\_conditions \leftarrow C(\pi)$$

✓ Proof analysis. Upon receiving the valid proof $\pi$, the adversary analyzes it to find any possible deviations from the expected format or content that could be used to justify rejection. The analysis could involve applying a polynomial-time algorithm $A$ that looks for any such deviations:

$$deviation \leftarrow A(\pi, challenge\_conditions)$$

✓ Arbitrary rejection criteria. The adversary may establish arbitrary or overly strict criteria for proof acceptance that are not part of the standard verification procedure. It might define a rejection algorithm $R$ that applies these criteria to the proof:

$$reject\_decision \leftarrow R(\pi, arbitrary\_criteria)$$

✓    Decision process. The adversary then decides to reject the proof based on the deviation found or the arbitrary criteria, even though $\pi$ is a valid proof for $S_{true}$:

$$output = \begin{cases} reject, \; if \; deviation \; or \; arbitrary \; criteria \; are \; meet \\ accept, \; otherwise \end{cases}$$

✓    Probability of incorrect rejection. The probability $Pr[Adversary \; rejects]$ that the adversary incorrectly rejects a valid proof is given by the following:

$$Pr[Adversary \; rejects] = Pr[output = reject \mid \pi \; is \; valid \; for \; S_{true}]$$

For a ZKP system that is complete, this probability should be 0, meaning that a valid proof for a true statement should always be accepted.

✓    Adversary's advantage.

The adversary's advantage $Adv_C[A]$ in the *completeness game* is the probability that they incorrectly reject a valid proof:

$$Adv_C[A] = Pr[Adversary \; rejects]$$

The *completeness property* of the ZKP system is violated if $Adv_C[A]$ is non-negligible. A secure and complete ZKP system should not allow any polynomial-time adversary to have a non-negligible advantage in this game, ensuring that valid proofs are always accepted by honest verifiers.

● Winning condition. The adversary wins if they reject a valid proof for $S_{true}$.

The adversary's goal is to reject valid proof. The winning condition is, thus, defined by the adversary's ability to find justification for rejecting proof that should be accepted according to the protocol. The *completeness property* stipulates that an honest verifier will always accept valid proof for a true statement.

✓    Verification decision. An honest verifier runs a verification algorithm $Verify$, which takes a proof $\pi$ and a statement $S$ and outputs a decision:

$$decision \leftarrow Verify(\pi, S)$$

The decision is a binary outcome where "accept" indicates a valid proof, and "reject" indicates an invalid proof.

✓    Proof acceptance criteria. For a proof $\pi$ to be accepted, it must meet the criteria defined by the verification algorithm, typically involving certain mathematical checks that correspond to the properties of the ZKP protocol.

✓    Adversary's rejection strategy. The adversary may apply an incorrect or non-standard verification strategy, denoted as $A$, that deviates from $Verify$ to find grounds for rejection:

$$incorrect\_decision \leftarrow A(\pi, S)$$

This strategy intentionally or erroneously rejects proofs that should be accepted under the correct verification process.

✓    Winning the game. The adversary wins if they can reject a proof $\pi$ that is valid for a true statement $S$ using their strategy:

$$Winning \; Condition = \{incorrect\_decision = reject \land decision = accept\}$$

✓    Probability of incorrect rejection. The probability $Pr[Adversary \; wins]$ that the adversary wins the game is the probability they reject a valid proof:

$$Pr[Adversary \; wins] = Pr[incorrect\_decision = reject \mid (\pi, S) \; is \; valid]$$

✓     Adversary's advantage:

The adversary's advantage $Adv_C[A]$ in the completeness game is quantified by the probability of them rejecting a valid proof:

$$Adv_C[A] = Pr[Adversary\ wins]$$

In a ZKP system that is complete, this advantage should be 0, indicating that no adversary can reject valid proof.

In the context of a secure ZKP system, the *completeness property* ensures that any valid proof of a true statement will be accepted by an honest verifier. Therefore, the adversary should not be able to win this game under normal circumstances; any strategy they employ that leads to a rejection of valid proof indicates a violation of the completeness property. The system is considered complete if the adversary's advantage $Adv_C[A]$ in rejecting valid proofs is negligible, ideally zero.

Advantage. The advantage is the probability that the adversary incorrectly rejects a valid proof, which should be negligible because, in a complete ZKP system, valid proofs are always accepted.

The advantage computation is summarized as follows:

- Let $Pr[Reject]$ be the probability that the adversary rejects a valid proof.
- Completeness dictates that a valid proof should always be accepted, so $Pr[Reject]$ should be 0 in a perfect system.
- The advantage $Adv_C[A]$ of the adversary in the *completeness game* is as follows:

$$Adv_C[A] = Pr[Reject]$$

For a complete system, $Adv_C[A]$ should be 0, meaning that the adversary never incorrectly rejects a valid proof.

Game 4. Attribute privacy game

- Goal. The adversary tries to determine specific attributes or the identity of the prover from a ZKP.
- Setup. The challenger generates a ZKP for a set of credentials that include private attributes.
- Attack. The adversary receives the ZKP and attempts to determine specific attributes or the identity of the prover.

The adversary's objective is to determine specific attributes or the identity of the prover from the ZKP. The attack algorithm in this context is aimed at extracting or deducing private information that should be concealed by the ZKP.

✓     Intercepting the proof. The adversary intercepts or is provided with a ZKP $\pi$, which proves a statement SS without revealing specific private attributes. The $\pi$ is generated based on the witness ww, which contains the private attributes.

✓     Analysis of the proof. The adversary applies an analysis algorithm $A$ to the proof $\pi$ in an attempt to extract information about the private attributes:

$$extracted\_info \leftarrow A(\pi)$$

This algorithm could involve statistical analysis, pattern recognition, or other cryptanalytic techniques.

✓     Deduction of attributes. The adversary tries to deduce the hidden attributes based on the information extracted from $\pi$. This could involve correlating the extracted data with known patterns or external information. The deduction can be formalized as a function $D$ that takes the extracted information and attempts to infer the private attributes:

$$inferred\_attributes \leftarrow D(extracted\_info)$$

✓     Comparison with known information. If the adversary has access to auxiliary information or a database of known attributes, they might compare the inferred attributes with this database to increase the accuracy of their deduction.

✓     Probability of successful inference. The probability $Pr[Adversary\ infers]$ that the adversary successfully infers private attributes is as follows:

$$\mathrm{Pr}[Adversary\ infers] = Pr[inferred\_attributes = actual\_attributes \mid extracted\_info \leftarrow A(\pi)]$$

In a secure ZKP system designed for attribute privacy, this probability should be negligible, meaning that the adversary should not be able to correctly infer the private attributes with any significant probability.

✓     Adversary's advantage. The adversary's advantage $Adv_{AP}[A]$ in the attribute privacy game is defined as their success rate in inferring the private attributes over random guessing:

$$AdvAP[A] = Pr[Adversary\ infers] - \frac{1}{number\ of\ possible\ attribute\ combinations}$$

If $Adv_{AP}[A]$ is non-negligible, the ZKP system fails to protect attribute privacy adequately. The goal of a secure system is to ensure that $Adv_{AP}[A]$ is negligible, thereby preserving the privacy of the prover's attributes.

●     Winning condition. The adversary wins if they can correctly identify any private attribute or the identity of the prover from the ZKP.

The winning condition for the adversary is to successfully infer specific private attributes or the identity of the prover from the ZKP, which should not be possible if the system effectively protects attribute privacy.

✓     Private attributes. Let $A$ represent the set of private attributes or the identity information that the prover aims to keep confidential while proving a statement $S$ with the ZKP. The adversary's goal is to infer information about these attributes from the proof $\pi$.

✓     Adversary's inference process. The adversary applies an attack algorithm $A$ to the proof $\pi$ to extract or deduce information about the private attributes:

$$inferred\_attributes \leftarrow A(\pi)$$

✓     Winning the game. The adversary wins the game if the inferred attributes match the actual private attributes $A$:

$$Winning\ Condition = \{inferred\_attributes = A\}$$

✓     Probability of correct inference.

The probability $Pr[Adversary\ wins]$ that the adversary wins the game is the probability that the inferred attributes match the actual attributes:

$$Pr[Adversary\ wins] = Pr[inferred\_attributes = A]$$

✓     Quantifying success. The success of the adversary in winning the game can be quantified as a measure of how accurately the adversary can guess or deduce the private attributes compared to a random guess.

If $\mid A \mid$ represents the number of possible attribute combinations, the adversary's success rate should not be significantly better than $\frac{1}{|A|}$, the probability of a correct guess by random chance.

✓     Adversary's advantage. The adversary's advantage $Adv_{AP}[A]$ in the attribute privacy game is defined as follows:

$$Adv_{AP}[A] = Pr[Adversary\ wins] - \frac{1}{|A|}$$

For a ZKP system to be considered secure in terms of attribute privacy, $Adv_{AP}[A]$ should be negligible. This means that the adversary's probability of successfully inferring the private attributes should be only marginally better, if at all, then what would be achieved by random guessing. A negligible $Adv_{AP}[A]$ indicates that the ZKP system effectively conceals the private attributes, ensuring the privacy of the prover's information.

Advantage. The adversary's advantage is the probability of successfully identifying a private attribute or identity minus the probability of a random guess.

The adversary's advantage in this game is defined as the ability to guess a hidden attribute better than random guessing would allow. Let us denote the hidden attribute as aa within the set of possible attributes $A$.

Advantage computation is summarized as follows:

- Let $Pr[Guess]$ be the probability that the adversary correctly guesses the attribute $a$.
- Let $\frac{1}{|A|}$ be the probability of a correct guess by random chance, where $\mid A \mid$ is the number of possible attributes.
- The advantage $Adv_{AP}[A]$ of the adversary in the *attribute privacy game* is as follows:

$$Adv_{AP}[A] = Pr[Guess] - \frac{1}{|A|}$$

In a secure ZKP system, $Adv_{AP}[A]$ should be negligible, meaning the adversary's probability of guessing correctly is not significantly better than random chance.

Analysis: This game tests the privacy-preserving property of ZKPs, ensuring that no private information can be inferred from the proof. A strong ZKP system should prevent the adversary from having any significant advantage in this game.

Game 5. Non-malleability game

- Goal. The adversary tries to modify a *ZKP* to create a new proof for a different statement or a different prover without detection.
- Setup. The challenger generates a *ZKP* for a statement $S$ with respect to a prover's credentials.
- Attack. The adversary receives the *ZKP* and attempts to alter it to create a new proof $ZKP'$ for a different statement $S'$ or to appear as if it came from a different prover.

The adversary's goal is to alter a valid ZKP to create a new proof for a different statement or to make it appear as if it came from a different prover. The attack algorithm in this context aims to modify or transform a given ZKP without detection.

- ✓ Original proof. The adversary obtains a valid ZKP $\pi$ for a statement $S$ and a public key $PK$. The $\pi$ is generated by an honest prover based on a witness $w$.
- ✓ Proof Modification. The adversary applies a modification algorithm $M$ to $\pi$ in an attempt to create a new proof $\pi'$ for a different statement $S'$ or to make it appear from a different prover:

$$\pi' \leftarrow M(\pi, S, S', modification\_parameters)$$

The modification parameters guide how $\pi$ is altered to fit $S'$ or a different prover's characteristics.

- ✓ Mimicking valid proof. The adversary tries to ensure that $\pi'$ maintains the appearance of a valid proof under the verification process for $S'$ or the new prover's public key. This might involve replicating the structure of a legitimate proof or cleverly disguising the modifications.
- ✓ Avoiding detection. The adversary needs to make sure that the modifications in $\pi'$ are not easily detectable by the standard verification process. This requires

a deep understanding of the verification algorithm and the ability to mimic its expected outputs.

✓ Probability of successful modification. The probability $Pr[Adversary\ modifies]$ that the adversary successfully modifies the proof is given by the following:

$$Pr[Adversary\ modifies] = Pr[Verifier\ accepts\ \pi'\ as\ valid\ for\ S'\ or\ new\ prover\ |\ \pi' \\ \leftarrow M(\pi, S, S', modification\_parameters)]$$

In a ZKP system with strong non-malleability, this probability should be negligible, meaning that the adversary should not be able to successfully modify the proof and have it accepted as valid with any significant probability.

✓ Adversary's advantage. The adversary's advantage $Adv_{NM}[A]$ in the non-malleability game is defined as the probability that they successfully create a malleable proof:

$$Adv_{NM}[A] = Pr[Adversary\ modifies]$$

A ZKP system is considered non-malleable if $Adv_{NM}[A]$ is negligible. This means that it is practically impossible for any adversary to modify a valid proof for one statement or prover and have it accepted as a valid proof for a different statement or prover.

● Verification. The challenger verifies $ZKP'$ to see if it is a valid proof for $S'$ or appears to be from a different prover.

The verification algorithm is used to assess whether a proof was tampered with or modified by an adversary. This algorithm aims to ensure that any alterations to a valid proof, intended to change its meaning or apparent origin, are detected.

✓ Original proof verification. Initially, the verifier has a standard verification algorithm $Verify$ that checks the validity of a proof $\pi$ for a statement $S$ under a public key $PK$:

$$output_{original} \leftarrow Verify(PK, S, \pi)$$

This output is either 'accept' or 'reject', indicating the proof's validity.

✓ Modified proof verification. The adversary presents a modified proof $\pi'$ for a possibly different statement $S'S'$ or under a different prover's public key. The verifier uses the same or an enhanced verification algorithm to assess $\pi'$:

$$output_{modified} \leftarrow Verify(PK', S', \pi')$$

$PK'$ could be the same or a different public key, depending on whether the adversary is attempting to change the statement or the apparent prover.

✓ Detection of modifications. The verification algorithm must be robust enough to detect any non-trivial modifications to $\pi$ that would alter its meaning or origin. This could involve additional checks for structural integrity, consistency with the public key, or other cryptographic properties that a valid proof must satisfy.

✓ Comparison with original proof. If possible, the verifier may also compare $\pi'$ with the original proof $\pi$ to identify any discrepancies or modifications:

$$comparison_{result} \leftarrow Compare(\pi, \pi')$$

✓ Algorithm complexity. The verification algorithm is efficient and runs in polynomial time relative to the size of the input to ensure practicality in real-world applications.

✓ Probability of detecting modifications. The probability $Pr[Detection]$ that the verifier detects the modifications is as follows:

$$Pr[Detection] = Pr\left[output_{modified} = reject\ |\ \pi'\ is\ a\ modified\ proof\right]$$

In a non-malleable ZKP system, this probability should be high, ideally close to 1, meaning that any significant modifications to a valid proof are almost always detected.

✓ Verification robustness. The robustness of the verification algorithm is essential in ensuring non-malleability. The algorithm must be sensitive enough to detect alterations that could change the meaning of the proof or its origin while still accepting valid, unaltered proofs. A high probability of detecting modifications while maintaining a low false rejection rate of valid proofs is indicative of a strong non-malleable verification system.

• Winning condition. The adversary wins if ZKP′ is accepted as valid for $S'$ or appears to be from a different prover despite the original proof being for $S$ and the original prover.

The winning condition for the adversary is to successfully modify a valid proof $\pi$ for a statement $S$ and have it accepted as a valid proof $\pi'$ for a different statement $S'$ or as if it came from a different prover. This winning condition challenges the non-malleability aspect of the ZKP, which is designed to prevent such alterations.

✓ Proof modification. The adversary modifies a valid proof $\pi$ to create a new proof $\pi'$. The modification is intended to change the statement being proven from $S$ to $S'$ or to make it appear as if $\pi'$ came from a different prover.

✓ Verification of the modified proof. The verifier uses their verification algorithm *Verify* to check the validity of $\pi'$:

$$output \leftarrow Verify\big(PK', S', \pi'\big)$$

$PK'$ is the public key associated with $S'$ or the new supposed prover.

✓ Winning the game. The adversary wins the game if the verification algorithm accepts the modified proof:

$$Winning\ condition = \{output = accept\}$$

✓ Probability of winning. The probability $Pr[Adversary\ wins]$ that the adversary wins the game is the probability that the verifier accepts the modified proof $\pi'$:

$$Pr[Adversary\ wins] = Pr\big[Verify\big(PK', S', \pi'\big) = accept \mid \pi'\ is\ modified\ from\ \pi\big]$$

✓ Non-malleability violation. If $Pr[Adversary\ wins]$ is non-negligible, then the adversary successfully violated the non-malleability property of the ZKP system. Non-malleability is considered breached if the adversary can feasibly modify a proof for one statement and have it accepted as valid proof for a different statement or prover.

✓ Adversary's advantage. The adversary's advantage $Adv_{NM}[A]$ in the non-malleability game is defined as their success rate in creating a malleable proof:

$$Adv_{NM}[A] = Pr[Adversary\ wins]$$

For a ZKP system to maintain its integrity and non-malleability, $Adv_{NM}[A]$ must be negligible. This ensures that any attempt to alter a valid proof to change its intended meaning or origin is highly unlikely to succeed, thus preserving the trustworthiness and reliability of the proof system.

Advantage. The adversary's advantage is defined as the probability of creating a valid ZKP′ for $S'$ or a different prover minus the probability of doing so by chance.

For non-malleability, the adversary's goal is to modify a proof to either change the statement being proved or the identity of the prover. We define the advantage based on their success in doing so.

Advantage computation can be summarized as follows:

• Let $Pr[Forge]$ be the probability that the adversary creates a new valid proof ZKP′ for a different statement $S'$ or a different prover.

- Since a valid ZKP should be non-malleable, the expected success probability for the adversary by random chance is negligible (denoted as $negl(n)$, where $n$ is the security parameter).
- The advantage $Adv_{NM}[A]$ of the adversary in the non-malleability game is as follows:

$$Adv_{NM}[A] = Pr[Forge] - negl(n)$$

In a ZKP system with strong non-malleability, any $PPT$ adversary's probability of creating a malleable proof should be negligible, so $Adv_{NM}[A]$ should also be negligible.

## 6. Conclusions

The key findings of the study involve the development of a novel authentication scheme that leverages the principles of Web 3.0, such as decentralization and user-centric control. This scheme aims to improve online security, privacy, and trust by enabling reliable identity verification without excessive disclosure of personal information. The paper concludes with an analysis of the security aspects of the proposed scheme, emphasizing its potential to transform digital identity management in a Web 3.0 environment.

The work emphasizes the significant change in digital identity due to the emergence of Web 3.0. This move is more than just a technological improvement; it involves a fundamental restructuring of online identity, providing many options to transform digital life.

Web 3.0 enables people to have more control over their digital identities through self-sovereign identification (SSI). Users can manage the sharing and utilization of their personal information. We observed that technologies such as blockchain offer improved security measures, including encryption and decentralization, enhancing the protection of digital identities against fraud and theft.

Digital identities in Web 3.0 can be created to be portable and interoperable, allowing them to work across many platforms and services, providing users with a more smooth and unified experience.

The advancement of digital identity in Web 3.0 allows for the development of innovative applications and services that utilize the greater capabilities of these identities, including better personalization, decentralized financing (DeFi), and other features.

Web 3.0 digital IDs, if deployed correctly, can enhance inclusiveness by offering identity solutions to individuals currently not well served by existing systems.

As proposed future developments, we plan to perform the following:

- Provide and make public the implementation once the patent is approved [179].
- Enhanced key management solutions. Develop user-friendly key management solutions that include secure key recovery options, such as social recovery, multi-signature schemes, or encrypted backup stored in a decentralized network.
- Hybrid authentication models. Integrate the proposed scheme with traditional authentication methods in a hybrid model, offering users multiple options for authentication. This approach can ease the transition for users and provide fallback options in case of key loss.
- Scalability improvements. Leverage advancements in distributed ledger technology, such as sharding or layer-2 solutions, to improve the scalability of the system. This could help manage the verification of credentials and signatures more efficiently on a scale.
- User education and onboarding. Implement comprehensive user education programs to simplify the onboarding process. Interactive tutorials, simulations, and customer support can help demystify the technology and encourage adoption.
- Zero-knowledge proof enhancements. Invest in research and development to enhance the efficiency and applicability of ZKPs. This could involve developing more general-purpose ZKP protocols that are easier to integrate into various applications.
- Interoperability standards. Work towards establishing interoperability standards for digital identities and credentials. This would facilitate seamless authentication across

different platforms and services, enhancing the user experience and expanding the system's applicability.

By focusing on these specific areas, the suggested authentication technique can be improved and positioned as a practical, secure, and user-friendly alternative to conventional authentication methods. This has the potential to be widely adopted in many applications and services.

**Author Contributions:** Methodology, S.L.N. and M.I.M.; Software, S.L.N. and M.I.M.; Validation, S.L.N. and M.I.M.; Writing—original draft, S.L.N. and M.I.M.; Writing—review & editing, S.L.N. and M.I.M. All authors have read and agreed to the published version of the manuscript.

**Funding:** This research received no external funding.

**Data Availability Statement:** Data are contained within the article.

**Conflicts of Interest:** The authors declare no conflicts of interest.

**Appendix A**

To assess user adoption and satisfaction within the context of the digital identity system described, a detailed survey can be conducted. This survey captures various aspects of user interaction with the system, including ease of use, security perceptions, privacy concerns, and overall satisfaction. The objective is to gather actionable insights that could guide improvements and increase user engagement. Here is an outline for such a survey:

User adoption and satisfaction survey

Section 1: Demographics

Age range:

- Under 18;
- 18–24;
- 25–34;
- 35–44;
- 45–54;
- 55–64;
- 65+.

Occupation:

- Student;
- Professional/Technical;
- Managerial/Business;
- Homemaker;
- Retired;
- Other (Please Specify).

Frequency of digital service use:

- Daily;
- Several times a week;
- Weekly;
- Monthly;
- Rarely.

Section 2: System use and utility How did you learn about our digital identity system?

- Social Media;
- Friend/Family Recommendation;
- Online Advertisement;
- News Article;
- Other (Please Specify).

How long have you been using our digital identity system?

- Less than a month;

- 1–6 months;
- 7–12 months;
- More than a year.

For what purposes do you use the digital identity system? (Select all that apply)

- Accessing online services;
- Verifying identity to third parties;
- Secure transactions;
- Other (Please Specify).

Section 3: Ease of Use and Interface How would you rate the ease of generating your public–private key pair using our application?

- Very easy;
- Somewhat easy;
- Neutral;
- Somewhat difficult;
- Very difficult.

How clear were the instructions for creating and managing your digital identity?

- Very clear;
- Somewhat clear;
- Neutral;
- Somewhat unclear;
- Very unclear.

How would you rate the overall user interface and experience?

- Excellent;
- Good;
- Average;
- Poor;
- Very poor.

Section 4: Security and privacy How confident are you in the security of your digital identity when using our system?

- Very confident;
- Somewhat confident;
- Neutral;
- Somewhat unconvinced;
- Very unconvinced.

How satisfied are you with the privacy measures in place for protecting your identity information?

- Very satisfied;
- Somewhat satisfied;
- Neutral;
- Somewhat dissatisfied;
- Very dissatisfied.

Have you ever encountered any security or privacy issues while using the system?

- Yes;
- No;
- If yes, please describe the issue(s):

Section 5: Overall satisfaction and feedback Overall, how satisfied are you with the digital identity system?

- Very satisfied;
- Somewhat satisfied;
- Neutral;
- Somewhat dissatisfied;

- Very dissatisfied.

What do you like most about the system? (Open text response) What improvements would you suggest for the system? (Open text response) Section 6: Future Use and Recommendations How likely are you to continue using our digital identity system in the future?

- Very likely;
- Somewhat likely;
- Neutral;
- Somewhat unlikely;
- Very unlikely.

How likely are you to recommend our digital identity system to others?

- Very likely;
- Somewhat likely;
- Neutral;
- Somewhat unlikely;
- Very unlikely.

Closing remarks

Thank you for participating in our survey. Your feedback is invaluable to us in improving our services and ensuring the best user experience possible. Please submit any additional comments or suggestions you may have.

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
