# Peer review of "A Novel Authentication Scheme Based on Verifiable Credentials Using Digital Identity in the Context of Web 3.0"

_electronics, doi:10.3390/electronics13061137_

Round 1

Reviewer 1 Report

Comments and Suggestions for Authors

1. In the literature review, the article cites a large number of references to discuss the research status. However, most of them are merely listed without elaborating on the strengths and weaknesses of the studies, making it difficult to serve as a basis for citing one's own opinion.

2. The clarity of Figure 1 is not optimal. It is recommended to insert a high-definition vector graphic instead.

3. The proposed solution is described as decentralized on page 24, line 1158 of the article. However, this aspect is not clearly illustrated in Figure 1 or in the accompanying textual descriptions.

4. The description and citation of relevant content in the article (e.g., page 9, line 402) are not formatted correctly, please refer to published papers for guidance on proper formatting.

Author Response

Dear Reviewer #1,

Thank you very much for your efforts in providing constructive and positive critiques of our work.

We have addressed each concern, comment, and question provided.

Also, some of the comments were quite volatile in terms of meaning, we have tried to do our best.

For some of the comments provided we had our point of view because we felt that our work was pulled down from the beginning process of submitting and following up with you as Reviewer.

Please see our answers to the comments and accept our apologies if our tone is inappropriate.

Hopping in a deep understanding with the situation,

You have the most respect and appreciation for the effort put into the review.

Best regards,

Marius Iulian Mihailescu

Reviewer 2 Report

Comments and Suggestions for Authors

This paper discusses the concept of digital identity in the context of Web 3.0 and proposes a novel authentication scheme using verifiable credentials. The paper provides a theoretical framework and explores the implications of this authentication scheme for user control, security, and privacy in digital interactions. The paper concludes by  discussing the broader implications of this scheme for future online transactions and digital identity management. But after careful review, I still have the following problems:

1. The paper cites a large number of literature and lists the content of the literature, but does not discuss, summarize and analyze it.

2. The legends accompanying figures require substantial improvement in their descriptions. Clear and concise legends are essential for readers to comprehend the presented data effectively. Clear and high-resolution figures are necessary.

3. Could you provide more implementation details for reproducing the proposed authentication scheme?

4. Have you conducted any evaluation or ablation studies to validate the effectiveness and robustness of the scheme? If so, could you provide more details on the experimental setup and results?

5. How does the proposed authentication scheme compare to existing authentication methods in terms of security, user experience, and scalability?

6.  English needs to be improved by correcting tense and grammatical errors.

7.  The structure and formatting of the paper needs greater clarity and accuracy:

(1) There is a problem with the indentation of line 1303.

(2) There is a space before the bold subtitle on line 1326. Please unify it.

(3) The title of line 1595, please confirm whether its logic is correct.

Comments on the Quality of English Language

English needs to be improved by correcting tense and grammatical errors.

Author Response

Dear Reviewer #2,

Thank you very much for your efforts in providing constructive and positive critiques of our work.

We have addressed each concern, comment, and question provided.

Also, some of the comments were quite volatile in terms of meaning, we have tried to do our best.

For some of the comments provided we had our point of view because we felt that our work was pulled down from the beginning process of submitting and following up with you as Reviewer.

Please see our answers to the comments and accept our apologies if our tone is inappropriate.

Hopping in a deep understanding with the situation,

You have the most respect and appreciation for the effort put into the review.

Best regards,

Marius Iulian Mihailescu

Reviewer 3 Report

Comments and Suggestions for Authors

1. The paper provides a general overview of the paper's focus on digital identity and the proposed authentication scheme. However, it lacks clarity regarding the specific contributions or innovations introduced by the research. It would be beneficial for readers if the abstract clearly articulated what novel aspects or advancements the paper brings to the field of authentication within the context of Web 3.0.

2. While the methods section mentions the utilization of blockchain, smart contracts, and cryptographic algorithms, it would be helpful to provide more detailed insights into how these technologies are integrated to form the proposed authentication scheme. A deeper explanation of the theoretical framework and the practical implementation of the scheme would enhance the paper's comprehensibility and applicability.

3. The paper briefly mentions the main findings, highlighting the proposed authentication scheme's ability to enhance user control, security, and privacy. However, it lacks information on the specific evaluation methodologies employed to assess these aspects. Providing details on how the scheme was tested, validated, or compared against existing methods would strengthen the credibility and reliability of the results.

4. While the conclusions discuss the broader implications of the proposed scheme for online transactions and digital identity management, they could benefit from further elaboration on potential real-world applications and scenarios where the scheme could be deployed. Additionally, outlining specific avenues for future research or development in this area would provide valuable insights for readers interested in advancing the field.

5. Overall, while the paper covers various aspects of the paper, the structure could be improved for better coherence and flow. Clarifying the sequence of information and ensuring each section of the paper is concise yet informative would enhance its readability and effectiveness in conveying the paper's key contributions and findings.

Comments on the Quality of English Language

-

Author Response

(The authors gave the same response as above.)

Round 2

Reviewer 1 Report

Comments and Suggestions for Authors

N/A

Reviewer 2 Report

Comments and Suggestions for Authors

The author responded point-to-point to my questions and revised the paper. The amount of revision work is enough to prove that the author takes it seriously. I don't have any more comments. Thanks to the authors and editors for their efforts.

Reviewer 3 Report

Comments and Suggestions for Authors

N/A

Comments on the Quality of English Language

N/A